# Changing effects of external forcing on Atlantic-Pacific interactions

Soufiane Karmouche[1,2], Evgenia Galytska[1,2], Gerald A. Meehl[3], Jakob Runge[4,5], Katja Weigel[1,2], and Veronika Eyring[2,1]

[1]University of Bremen, Institute of Environmental Physics (IUP), Bremen, Germany
[2]Deutsches Zentrum für Luft- und Raumfahrt e.V. (DLR), Institut für Physik der Atmosphäre, Oberpfaffenhofen, Germany
[3]Climate and Global Dynamics Laboratory, National Center for Atmospheric Research (NCAR), Boulder, CO, USA
[4]Deutsches Zentrum für Luft- und Raumfahrt e.V. (DLR), Institut für Datenwissenschaften, Jena, Germany
[5]Fachgebiet Klimainformatik, Technische Universität Berlin, Berlin, Germany

**Correspondence:** Soufiane Karmouche (sou_kar@uni-bremen.de)

**Abstract.** Recent studies have highlighted the increasingly dominant role of external forcing in driving Atlantic and Pacific Ocean variability during the second half of the 20[th] century. This paper provides insights into the underlying mechanisms driving interactions between modes of variability over the two basins. We define a set of possible drivers of these interactions and apply causal discovery to reanalysis data, two ensembles of pacemaker simulations where sea surface temperatures in either the tropical Pacific or the North Atlantic are nudged to observations, and a pre-industrial control run. We also utilize large ensemble means of historical simulations from the Coupled Model Intercomparison Project Phase 6 (CMIP6) to quantify the effect of external forcing and improve the understanding of its impact. A causal analysis of the historical time series between 1950 and 2014 identifies a regime switch in the interactions between major modes of Atlantic and Pacific climate variability in both reanalysis and pacemaker simulations. A sliding window causal analysis reveals a decaying ENSO effect on the Atlantic as the North Atlantic fluctuates towards an anomalously warm state. The causal networks also demonstrate that external forcing contributed to strengthening the Atlantic's negative-sign effect on ENSO since the mid-1980s where warming tropical Atlantic sea surface temperatures induce a La Niña-like cooling in the equatorial Pacific during the following season through an intensification of the Pacific Walker circulation. The strengthening of this effect is not detected when the historical external forcing signal is removed in the Pacific pacemaker ensemble. The analysis of the pre-industrial control run supports the notion that the Atlantic and Pacific modes of natural climate variability exert contrasting impacts on each other even in the absence of anthropogenic forcing. The interactions are shown to be modulated by the (multi)decadal states of temperature anomalies of both basins with stronger connections when these states are "out of phase". We show that causal discovery can detect previously documented connections and provides important potential for a deeper understanding of the mechanisms driving changes in regional and global climate variability.

## 1 Introduction

One of the biggest challenges in climate science is associated with disentangling the effects of internal climate variability and external forcings and robust quantification of their impacts. External forcing includes changes to the climate system caused by natural factors such as volcanic eruptions, solar radiation, or from the human emission of greenhouse gas (GHG) and

aerosols. Internal climate variability, on the other hand, refers to the inherent fluctuations in the climate system that arise from complex interactions between the atmosphere, oceans, land, and cryosphere (Masson-Delmotte et al., 2023). For instance, the interactions between the ocean and atmosphere produce unique patterns of variability at different time scales, such as El Niño-Southern Oscillation (ENSO, Bjerknes, 1966; BJERKNES, 1969; Neelin et al., 1998), Pacific Decadal Variability (PDV, Mantua et al., 1997; Newman et al., 2016), and Atlantic Multidecadal Variability (AMV, Trenberth and Shea, 2006; Zhang et al., 2019). Despite the considerable advancements through different phases of the Coupled Model Intercomparison Project (CMIP), including the ability to represent the statistical properties and spatial patterns of modes of climate variability, accurately simulating them as in the observed record still poses daunting challenges (Eyring et al., 2019; Fasullo et al., 2020; Karmouche et al., 2023). In addition to the short historical record, the intricate nature of the climate system and the non-linear characteristics of natural climate variability impose inherent limitations on predictability and introduce inseparable uncertainty into climate model projections (Deser et al., 2010; Meehl et al., 2013; Fasullo et al., 2020; Eyring et al., 2021; Deser and Phillips, 2023). This underlines the importance of studying natural climate variability modes and their teleconnections to advance our understanding of the climate system and improve the reliability of climate model projections (Deser and Phillips, 2023).

Previous studies have explored connections between the Atlantic and Pacific basins, focusing on the influence of the Walker circulation (BJERKNES, 1969) on both interannual and decadal timescales (Latif and Grötzner, 2000; McGregor et al., 2014; Meehl et al., 2016; Ruprich-Robert et al., 2017; Meehl et al., 2020). Other studies found that the Atlantic drives the Pacific, while the Pacific can also drive the Atlantic, and there are influences between the two through the tropical Indian Ocean (Kumar et al., 2013; Li et al., 2015; Ruprich-Robert et al., 2017; Levine et al., 2017; Meehl et al., 2016; Yang et al., 2019; An et al., 2021). For example, on the decadal timescale, Meehl et al. (2020) demonstrate how a positive AMV can lead to a negative PDV through anomalous Walker circulation. They showed that a negative PDV, which triggers a same-sign response in the tropical Atlantic, contributes to AMV's transition from a positive to a negative phase (Fig. 3 in Meehl et al., 2020). The study highlights that in addition to the tropical Walker circulation, positive convective heating and precipitation anomalies in the tropical Pacific can establish connections to extra-tropical modes of atmospheric variability, e.g. the Pacific-North American pattern (PNA) and Pacific South American pattern (PSA), which also contribute to the same-sign effect of the Pacific on the Atlantic. The tropical pathway connecting the Pacific and the Atlantic happens mainly through modifications to the Walker circulation affecting large-scale tropical weather systems. On the other hand, the extratropical southern hemisphere (through the PSA) and northern hemisphere teleconnections illustrated through the PNA and North Atlantic Oscillation (NAO, Barnett, 1985; Brönnimann et al., 2006; Scaife et al., 2014) not only connect the Atlantic-Pacific basins but also play important roles in modulating Arctic sea ices responses (Polyakov and Johnson, 2000; Meehl et al., 2018; Galytska et al., 2022) and shaping European climate (Brönnimann et al., 2006; Brönnimann, 2007).

Similar to the connection between AMV and PDV, previous research revealed well-established contrasting responses between the Atlantic and Pacific tropical sea surface temperature (SST) also on the interannual timescale (Zebiak, 1993; Latif and Grötzner, 2000; Ham et al., 2013b; Martín-Rey et al., 2014a). At the equator, ENSO contributes to the Atlantic Zonal Mode (AZM), also known as the Atlantic Niño, through the Walker circulation (Saravanan and Chang, 2000; Sutton et al.,

2000; Wang, 2019). On the other hand, it is suggested that an Atlantic Niño, peaking during the boreal summer upwelling sea-
son, can induce a La Niña event in the Pacific during the following winter through an atmospheric bridge altering the Walker
circulation (Ham et al., 2013b, a; Martín-Rey et al., 2014a; Keenlyside et al., 2013). ENSO is also related to the second mode
of tropical Atlantic variability, known as the Atlantic Meridional Mode (AMM), expressed as a cross-equatorial SST gradient
with opposite signs between the tropical South Atlantic (TSA) and the tropical North Atlantic (TNA, Rajagopalan et al., 1998;
Enfield et al., 1999; Chiang and Vimont, 2004). Both AZM and AMM are linked to changes in the Intertropical Convergence
Zone (ITCZ) and associated winds (Masson-Delmotte et al., 2023). Several studies show that ENSO affects north tropical At-
lantic SST anomalies (SSTAs) during the spring and summer through tropical and extratropical pathways (Enfield and Mayer,
1997; Klein et al., 1999; Saravanan and Chang, 2000; Taschetto et al., 2015; Meehl et al., 2016; García-Serrano et al., 2017b;
Park et al., 2023b; Meehl et al., 2020). Conversely, SST variability in the tropical Atlantic can also influence an ENSO event
during the following boreal winter by modifying anomalous low-level zonal winds through atmospheric teleconnections over
the equatorial western Pacific (Ham et al., 2013b; Park and Li, 2018; Park et al., 2023b). In addition to ENSO, NAO also
contributes to forcing SSTAs in the TNA region (Czaja et al., 2002; Visbeck et al., 2003). The weak intensity of the subtropical
high during a negative NAO phase weakens the northeasterly winds, which favors TNA warming. The high SSTAs in the TNA
region trigger an atmospheric response that strengthens the connection between TNA and ENSO (Wang et al., 2017b; Park
et al., 2019). The impact of the Atlantic on ENSO, which has become more significant since the mid-1980s is associated with
an increase in the climatological mean SST in the North Atlantic, which can be attributed to a positive phase shift of AMV
and/or human-caused warming (Park and Li, 2018; Yang et al., 2021; Meehl et al., 2020; Park et al., 2023b). In turn, the effect
of ENSO on TNA has decreased since the mid-1980s, supporting the existence of two regimes of contrasting responses as pro-
posed by Meehl et al. (2020) and described by Park et al. (2023b) as a Pacific-driven regime (1950 to mid-1980s) as opposed
to an Atlantic-driven regime (from mid-1980s to 2014), where both span over multiple decades.

While debate over the precise attribution of the early-2000s warming slowdown and the mid-1980s regime switch is still
ongoing, it is evident that signals from natural internal variability, GHG-induced warming, and aerosol-cooling all play roles
in the observed changes in global temperature variability over the historical record (Brönnimann, 2007; Meehl et al., 2013;
Kosaka and Xie, 2013; Dong et al., 2014; Mann et al., 2014; McGregor et al., 2014; Li et al., 2015; Meehl et al., 2016;
Kucharski et al., 2015; Smith et al., 2016; Dong and McPhaden, 2017; Haustein et al., 2019). There is growing evidence,
however, that in the Atlantic, external forcing is responsible for the recent AMV changes and its widespread impacts (Murphy
et al., 2017; Klavans et al., 2022; He et al., 2023). In addition to that, limited coverage of the in-situ observations over the
tropical Pacific contributes to a lack of consensus among previous studies on the reasons behind the recently observed 1979-
2014 strengthening of the Walker circulation (Vecchi and Soden, 2007; Power and Kociuba, 2011; L'Heureux et al., 2013;
DiNezio et al., 2013; Kociuba and Power, 2014). Findings from Chung et al. (2019) indicate that internal variability linked
to the Interdecadal Pacific Oscillation (IPO, similar to PDV) likely played the dominant role in the recent strengthening of
the Walker circulation. Other studies, however, emphasize the role of SST variability in neighboring ocean basins, such as the
Atlantic (Kucharski et al., 2011; McGregor et al., 2014) and/or the Indian Ocean (Luo et al., 2012), in modulating the tropical
Pacific variability. This means that the anthropogenic contribution to the Atlantic SST warming during the recent decades

(Watanabe and Tatebe, 2019; Klavans et al., 2022) might have influenced the Pacific variability and its effect on the recent strengthening of the Walker circulation.

To address the effects of a changing climate on the interactions between these modes, it is necessary to isolate internal variability from external forcing. While the "signal-to-noise paradox" in climate models is still a topic of debate (Scaife et al., 2014; Wang et al., 2017a; Sato et al., 2018; Smith et al., 2019; Chylek et al., 2020; Klavans et al., 2021, 2022), the use of large ensemble simulations has been proven to be extremely helpful to capture the observed trends and to help the detection and attribution of anthropogenic climate change in the observational record (Meehl et al., 2013; Menary et al., 2020; Borchert et al., 2021; Deser, 2020; Tebaldi et al., 2021; Klavans et al., 2022; Deser and Phillips, 2023). Studies proved that large ensembles can provide a robust sampling of models' internal variability and help assess externally forced changes in the characteristics of simulated internal variability (Menary et al., 2020; Borchert et al., 2021; Klavans et al., 2022; Deser and Phillips, 2023). Therefore, this paper utilizes large ensembles from CMIP Phase 6 historical simulations (CMIP6, Eyring et al., 2016) to represent all historical natural and human-induced external forcing.

Beyond mere correlation, causal discovery aims to learn the underlying causes and effects of the climate system (Runge et al., 2023). Similar to our previous research (Karmouche et al., 2023), in this study we apply a causal discovery algorithm to understand the regime-dependent causal networks connecting coupled and atmospheric modes of climate variability over the Atlantic and Pacific. While Karmouche et al. (2023) addressed teleconnections happening at yearly-interannual to decadal lags, this paper focuses on the seasonal timescale. The study encompasses a causal analysis of reanalysis data to investigate the distinctive regimes of teleconnections during two historical periods as suggested by Meehl et al. (2020); Park et al. (2023b), namely the Pacific-driven regime (1950-1983) and the Atlantic-driven regime (1983-2014). Pacemaker simulations are also used to explore the roles of ENSO and North Atlantic SSTAs in the cross-basin interactions before and after removing the externally forced signal. Finally, to show that Atlantic-Pacific interactions happen naturally, even in the absence of external human influences, we show results using a pre-industrial control run.

## 2  Data

### 2.1  Observational and reanalyses datasets

To calculate SST-based indices over the Atlantic and Pacific during the observed historical 1950-2014 period, we use the Hadley Centre Sea Ice and Sea Surface Temperature (HadISST, Rayner et al., 2003) dataset. We also use sea level pressure (SLP) and zonal wind component (U) at 925 hPa obtained from the National Center for Environmental Prediction-National Center for Atmospheric Research reanalysis 1 dataset (NCEP-NCAR-R1 Kalnay et al., 1996) to calculate indices for the atmospheric modes (NAO and PNA) and the Pacific Walker circulation (PWC).

## 2.2 Pacemaker simulations

To address the effect of ENSO on the Atlantic while controlling for external forcing, we utilized a 10-member ensemble of the Community Earth System Model 2 (CESM2; Danabasoglu et al., 2020) in which SSTAs in the tropical Pacific (15S-15N) were nudged to National Oceanic and Atmospheric Administration (NOAA) Extended Reconstruction Sea Surface Temperature version 5 (ERSSTv5) data during 1880-2019. Similar to the reanalyses datasets (see Sect. 2.1), the analysis of Pacific pacemaker simulations focuses on the period from 1950 to 2014. To preserve the mean state and biases of the model, the SST nudging was only applied to the anomalies, not the total SST. The ensemble includes all CMIP6 time-varying external, natural, and anthropogenic forcings, using historical forcings prior to 2014 and SSP3-7.0 forcing thereafter. We extract surface temperature (TS), SLP, and U variables from this dataset.

Similar to the CESM2 Pacific pacemaker ensemble, we utilized a 10-member ensemble of CESM1 simulations where observations (ERSSTv3b) were used to nudge time-evolving SST anomalies in the North Atlantic (5-55°N, with a linearly tapering buffer zone extending to the equator and 60°N, covering the Atlantic basin). This Atlantic pacemaker ensemble runs from 1920 to 2013, but only data for the 1950-2013 period are used. The ensemble includes all CMIP5 time-varying external, natural, and anthropogenic forcings (Yang et al., 2020).

## 2.3 Pre-industrial control run

To further examine the interactions between the Atlantic and Pacific modes of internal variability under unforced scenarios, we use 250 years from the CESM2 pre-industrial control run (representative of the period prior to 1850, Eyring et al., 2016) to extract monthly averages for the same variables as in the pacemaker simulations and reanalysis (TS, SLP, U). We use the CESM2 model for its remarkable simulation of ENSO characteristics (Danabasoglu et al., 2020; Capotondi et al., 2020; Chen et al., 2021) and also to facilitate the comparison with the pacemaker ensemble results.

## 2.4 Indices

We calculate the indices for TNA, Niño3.4, PNA, NAO, and the PWC with respect to 1950-2014 climatology as follows:

- TNA (Tropical North Atlantic) Index is the area-weighted monthly SSTAs over the North Tropical Atlantic region 5.5–23.5°N, 58°–15°W (Enfield et al., 1999).

- Niño3.4 Index is the area-weighted monthly SSTAs over the equatorial pacific region 5°N–5°S, 170°–120°W (Trenberth, 1997).

- PNA (Pacific North American) Index is the leading EOF of (3-monthly averaged and area-weighted) SLP anomalies over the Pacific North America region 20–85°N, 120°E–120°W (Wallace and Gutzler, 1981).

- NAO (North Atlantic Oscillation) Index is the leading EOF of (3-monthly averaged and area-weighted) SLP anomalies over the North Atlantic region 20-80°N, 90°W-40°E (Hurrell and Deser, 2009).

- $PWC_u$ (Pacific Walker Circulation u component) as an index for the PWC and is defined as the monthly zonal wind anomaly at 925 hPa (or nearest available level for pacemaker simulations) over the equatorial pacific region (6°N–6°S, 180°-150°E, following Chung et al., 2019), where negative (positive) values indicate anomalous easterly (westerly) winds that imply a strengthening (weakening) of the PWC.

- ATL3 (equatorial Atlantic) Index is the area-weighted monthly SSTAs over the equatorial Atlantic region 3°N–3°S, 20°W–0° (Zebiak, 1993).

We also use indices for AMV and PDV to illustrate the decadal imprint of internal variability over the Atlantic and Pacific, a proxy for the long-term physical state of the two basins. For the causal analysis of the pre-industrial control run, the low-pass filtered versions of these indices are used to filter out composites of the timeseries depending on the different in-phase and out-of-phase combinations of AMV and PDV. Here, the AMV and PDV indices are calculated as follows:

- AMV (Atlantic Multidecadal Variability) Index (sometimes referred to as the AMO (Atlantic Multidecadal Oscillation) index) is defined as monthly SSTAs averaged over the North Atlantic region 0–60°N, 80–0°W (Trenberth and Shea, 2006).

- PDV (Pacific Decadal Variability) Index (sometimes referred to as the PDO (Pacific Decadal Variability) index) is defined as the standardized principal component (PC) time series associated with the leading EOF of area-weighted monthly SSTAs over the North Pacific region 20–70°N, 110°E–100°W (Mantua et al., 1997).

## 3 Methodology

### 3.1 Separating internal variability from the externally forced components

To isolate internal variability from the Pacific pacemaker simulations, we first calculate a multi-ensemble mean (MEM) for each variable, representing an estimate of the externally forced component. This is done by averaging three CMIP6 historical large ensemble means (with different numbers of ensemble members): CESM2 (11 members), MIROC6 (50 members), and UKESM1-0-LL (16 members). These models were chosen to represent the MEM, since they realistically simulate the spatiotemporal characteristics of the major modes of climate variability (notably ENSO and North Atlantic SST modes) during the historical period (Phillips et al., 2020; Fasullo et al., 2020; Karmouche et al., 2023). Moreover, to detect changes in various climate phenomena, the required number of members in an ensemble simulation may differ. Forced changes in ocean heat content can be detected with only a few members, while changes in atmospheric circulation or extreme precipitation and temperature may need 20-30 members (Deser et al., 2010; Tebaldi et al., 2021; Smith et al., 2022). Detecting forced changes in the characteristics of internal variability, such as its amplitude, spatial pattern, and remote teleconnections, may require even larger ensembles (Milinski et al., 2020; Smith et al., 2022; O'Brien and Deser, 2023; Deser and Phillips, 2023). The idea behind estimating the forced response from three different large ensembles with different numbers of realizations is to reduce any biases originating from the model's own representation of CMIP6 forcing and/or from the ensemble size. Because each of the

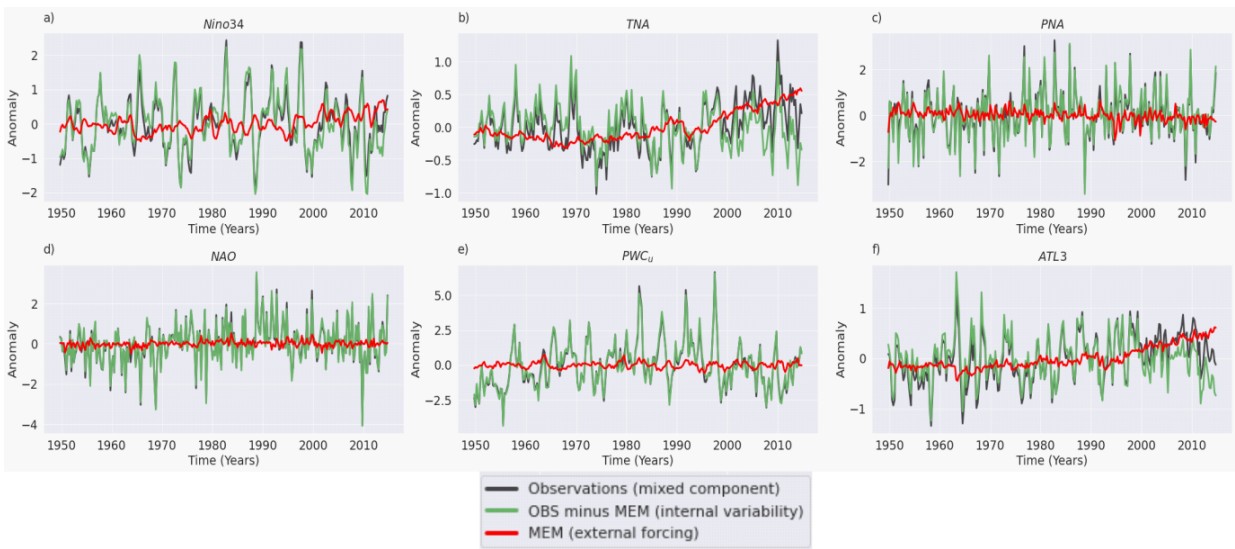

**Figure 1. Standardized seasonally averaged time series of a) Niño3.4, b) TNA, c) PNA, d) NAO, e) $PWC_u$, and f) ATL3 during the observed 1950-2014 period. Unit for the standard deviations are [°C] in a, b and f, [Pa] in c and d, and [m.s$^{-1}$] in e. The time series in black represent the mixed signal from indices calculated using HadISST (a,b,f) and NCEP-NCAR Reanalysis-1 (c-e). The time series in green are calculated after subtracting the CMIP6 external forcing represented by MEM following Eq. 1 (see Sect. 3.1). In each panel, the red line denotes the difference between the black and green line at each time step, representing the varying effect of subtracting MEM on each index.**

10 members in the pacemaker ensemble is subjected to the same CMIP6 time-varying external forcing, we assume that MEM is the response to external forcing such as GHG-induced warming trends, solar radiation, volcanic activity, land use changes, and anthropogenic aerosols. Consequently, the discrepancies in each pacemaker simulation relative to MEM can be attributed to internal variability. Therefore, for a given variable $X$ ($X = SST, U, SLP$), we can express the separation in a pacemaker simulation *i* as:

$$X_i = X_{\text{MEM}} + X_{\text{internal}(i)}, \qquad i = 1, 2, \ldots, 10 \tag{1}$$

where $X_{\text{MEM}}$ is the forced component estimated from the CMIP6 MEM and $X_{\text{internal}(i)}$ is the residual of the original $X_i$ minus the forced response $X_{\text{MEM}}$, which varies among different members and shows the component associated with isolated internal variability. This is similar to methods from Wu et al. (2021), but using the CMIP6 MEM instead of the pacemaker ensemble mean to quantify the forced component. It should be noted that the differences between the pacemaker simulations and MEM can emerge from: i) MEM estimating a climate sensitivity (in response to external forcing) different from the one prescribed in the restoring region. ii) Outside the restoring region, the climate sensitivity of the CESM models can also be different from the one estimated by MEM there. iii) Differences can also arise from a mix of (i) and (ii).

To isolate the internal variability in observations and the pacemaker simulations following Eq. (1), we subtract the MEM for SST, SLP, and U from observations and each pacemaker simulation before calculating the indices above. Figure 1 illustrates the

standardized seasonally averaged time series of Niño3.4 (a), TNA (b), PNA (c), NAO (d), $PWC_u$ (e), and ATL3 (f) indices from observations for the 1950-2014 period. For each index, the time series in black represents the original indices from HadISST (a,b,f) and NCEP-NCAR-R1 (c-e), as mixed signals including external forcing (non-linear trends). The time series in green show the indices after subtracting MEM (producing an estimate of isolated internal variability). The difference that denotes the effects of external forcing is shown in red. Based on Fig. 1, three indices expose positive trends represented by increased external forcing during the second half of the analyzed period, namely Niño3.4 (a) and more clearly TNA (b) and ATL3 (f). The rest of the analyzed indices do not show distinctive trends nor significant effects of external forcing.

## 3.2  PCMCI+ algorithm for causal discovery

Similar to methods from Karmouche et al. (2023), we use the PCMCI+ causal discovery algorithm, part of the freely available Tigramite python package (https://github.com/jakobrunge/tigramite, last access: 29.04.2024) to efficiently estimate causal networks from time series datasets (Runge et al., 2019), and detect lagged ($\tau$>0) and contemporaneous ($\tau$=0) causal links (Runge, 2020), where $\tau$ stands for the time lag. The full description of the method and pseudo code, along with explanations of the underlying assumptions for causal interpretation, can be found in Runge et al. (2023) and Runge (2020), respectively. Crucially, to interpret obtained links in PCMCI+ as causally directed, the method assumes that no unobserved confounders exist.

The PCMCI+ algorithm consists of two main phases: skeleton discovery and orientation. The skeleton discovery phase starts with the PC1 Markov set discovery algorithm, which is based on the PC algorithm (Peter Spirtes and Clark Glymour), to test for conditional independence of pairs of variables, and it is followed by a momentary conditional independence (MCI) test to remove spurious links due to contemporaneous confounders (Runge, 2020). The orientation phase orients contemporaneous links based on unshielded triples, and the resulting graph contains directed lagged and contemporaneous links as well as unoriented contemporaneous adjacencies (Markov equivalence) or conflicting adjacencies (see rules R1–R3 in Runge, 2020). We primarily apply partial correlation *ParCorr* as an independence test for the analysis, except in the proof of concept (Section 3.2.1) where we opt for *RobustParcorr*. *ParCorr* assesses partial correlation by regressing out the effect of a conditioning variable through linear ordinary least squares regression. While *RobustParcorr* is similar, it first transforms variables to the standard normal marginals which is particularly useful when dealing with non-gaussian distributed variables (which is the case for variables in Sect. 3.2.1; see skewed data in density plots in the Supplementary Fig. S1).

There are two main free parameters for PCMCI+. First, the maximum time lag $\tau_{max}$, which is decided after analyzing the lagged dependencies between the variables (see lag function plot in Supplementary Fig. S2) and literature review. The second parameter is $\alpha_{pc}$, which represents the significance threshold adopted for all PCMCI+ tests. The algorithm outputs a *p-matrix* (containing the *p-values*, denoting the uncertainty of each link) and a *val_matrix* (containing the *cross-MCI*, translating the strength of each link). The *p-values* for the coefficients shown on a PCMCI+ causal graph are below the significance threshold $\alpha_{pc}$. We note that these are valid only for the adjacencies and not the directionality of contemporaneous links decided during the orientation phase (lagged links are always oriented according to time order). The limitation presented by the absence of comprehensive confidence measures for all PCMCI+ estimated links is an aspect currently being addressed where bootstrap aggregation methods are still under review (e.g. Debeire et al., 2023). In the context of this paper, the *p-value* matrices are

shown in Supplementary material (e.g. Figs. S3, S6, S8) to provide a measure of the uncertainties of the estimated adjacencies. The Tigramite package offers the ability to plot results in the form of a causal graph where nodes represent the time series

associated with each variable. In these graphs, the node color shows the auto-MCI value (auto-correlation i.e. self links) and the link color indicates the cross-MCI value (i.e. link strength) with blue indicating opposite-sign (negative) inter-dependency and red indicating same-sign (positive) inter-dependency. The link-associated time lags are shown as small labels on the curved links. If a link is detected at different lags, the indicated lags are sorted by link strength (i.e. by the absolute cross-MCI value). Contemporaneous links (at lag zero) are represented by straight lines. In the context of causal links between variables $X_i$

and $X_j$ at time $t$, the possible link types considered for any time lag ( $\tau$ ) are non-adjacent links (i.e., the pair is not directly connected) and direct links from $X_i$ at time $t - \tau$ to $X_j$ at time $t$ ($X_i^{t-\tau} \rightarrow X_j^t$). For $\tau = 0$ additional possible link types are opposite direct links ($X_i^t \leftarrow X_j^t$), unoriented links ($X_i^t \circ - \circ X_j^t$), and conflicting links ($X_i^t \times - \times X_j^t$) which can occur due to finite sample effects or violations of assumptions. While we tolerate the presence of unoriented adjacencies due to Markov equivalence ($X_i^t \circ - \circ X_j^t$, hereafter denoted by "o-o" symbol) in the causal graphs of the upcoming sections, we introduce

assumptions for the cases where conflicting links ($X_i^t \times - \times X_j^t$) occur. For the analysis of the Pacific and Atlantic pacemaker ensembles (Sect. 4.2), we introduce these assumptions on the basis of which variables have been nudged toward observed values. For example, in the PCMCI+ tests on the pacific pacemaker simulations, the implemented assumption states that if the method detects a contemporaneous adjacency between Niño3.4 and any other variable $X$, then the link should be directed from Niño3.4 to $X$ ($Ni\tilde{n}o3.4 \rightarrow X$). This is because Niño3.4 is the pacemaker as the SSTA is nudged towards observations in that region.

We follow the same approach for the Atlantic pacemaker, where the assumptions presume only outgoing contemporaneous links from TNA (as the SSTAs in that region are nudged to observations). Additionally, some other assumptions are introduced to not estimate specific links (having no physical basis according to well-established literature) or to overcome specific cases of "$\times - \times$"-type links by presuming their orientation according to background knowledge (well-defined physics from previous studies) and/or sensitivity tests (using different conditional independence tests, different $\alpha_{pc}$ and/or sliding window analysis).

While no assumption is predefined in next section's proof of concept, the assumptions introduced in each of the Results subsections (Sects. 4.1-4.3) are explained in detail at the end of Sect. 3.2.2. In Supplementary material, we show all original PCMCI+ graphs that possibly contain conflicting edges ($X_i^t \times - \times X_j^t$) and where all dependencies between all variables at all lags are considered (i.e. no assumptions introduced).

### 3.2.1 Proof of Concept: the 1997/1998 El Niño

The ENSO phenomenon has been a subject of intense scientific interest due to its profound impacts on global climate patterns. Among the ENSO events, the El Niño of 1997/1998 stands out as one of the most powerful and influential episodes in recorded history. The processes and the feedbacks involved with this event are relatively well understood and documented (e.g. McPhaden, 1999; Lengaigne et al., 2003). Thus, as a proof of concept for our methodology, we apply causal discovery to analyze this event and to physically interpret it in the context of well-known processes. This will provide insight to point

toward the causal analysis in this paper of the less well-understood interactions between the Pacific and the Atlantic. Through the application of the PCMCI+ causal discovery algorithm, we identify potential cause-and-effect relationships among the

selected variables, shedding light on the intricate interactions that contributed to the onset and intensification of the 1997/1998 El Niño event. Candidate variables for this case study, which have been used in previous studies (Trenberth, 1997; Neelin et al., 1998; McPhaden, 1999; Lengaigne et al., 2003; Wang, 2018) and thus applied here, include the SSTAs over the Niño3.4 region (SST Niño3.4), eastward wind anomalies in the central Pacific (Uwind CPAC), the wind stress over the west Pacific ($Wind_{Stress}$ WPAC), the east-west SLP anomaly gradient ($SLP_{grad}$ EPAC-WPAC), and the depth of thermocline in the east Pacific ($Tcline_{Depth}$ EPAC). We extract these variables as monthly averages between January 1995 and December 1999 from NCEP-NCAR-R1, HadISST, and two reanalysis datasets from the European Centre for Medium-Range Weather Forecasts (ECMWF), namely, the Ocean Reanalysis System 5 (ORAS5, Copernicus Climate Change Service, 2021) and ERA5 (Copernicus Climate Change Service, 2019) datasets. This means that the PCMCI+ data frame in this section has a length of 60 time steps (months) with 5 variables, which are listed in Table 1 with their respective details.

**Table 1.** Climate variables used in the 1997/1998 El Niño case study

| Variables (nodes) | Dataset | Definition | Region |
|---|---|---|---|
| SST Niño3.4 | HadISST | SSTAs over the Niño3.4 region [$°C$] | 5°S-5°N, 170°W-120°W |
| Uwind CPAC | NCEP-NCAR-R1 | Westerly Wind Anomalies in Central Pacific [$m.s^{-1}$] | 5°S-5°N, 180°W-150°W |
| $Wind_{Stress}$ WPAC | ORAS5 | West Pacific Wind Stress Anomalies [$N.m^{-2}$] | 5°S-5°N, 140°E-170°E |
| $SLP_{grad}$ EPAC-WPAC | NCEP-NCAR-R1 | East-West Sea Level Pressure Anomaly Gradient [$Pa$] | [5°S-5°N, 100°E-160°E] minus [5°S-5°N, 100°W-160°W] |
| $Tcline_{Depth}$ EPAC | ORAS5 | Depth of 20°C Isotherm in the Eastern Pacific [$m$] | 5°S-5°N, 150°W-120°W |

During this proof of concept section, we estimate dependencies only for lags between 0 and 3 months ($\tau_{min} = 0$, $\tau_{max} = 3$) and set the significance threshold $\alpha_{pc}$ to 0.1. The parameter $\tau_{max}$ is generally decided after inspecting a lagged dependency matrix (see Supplementary Fig. S2). The PCMCI+ algorithm detected causal links between the variables (see time series in Fig. 2a) that can be summarized through the causal network shown in Fig. 2b. The uncertainties for all coefficients in Fig. 2b can be found in the Supplementary Fig. S3.

The results obtained from the PCMCI+ algorithm confirm previous insights into the mechanisms leading to the 1997/1998 El Niño event and its intensification by identifying significant causal links among the variables. One crucial node with only outgoing links is $Wind_{Stress}$ WPAC, indicating the central role of the March 1997 westerly wind bursts in triggering the 1997/1998 El Niño, by generating a downwelling eastward-propagating Kelvin wave, as evidenced by previous studies (Lian

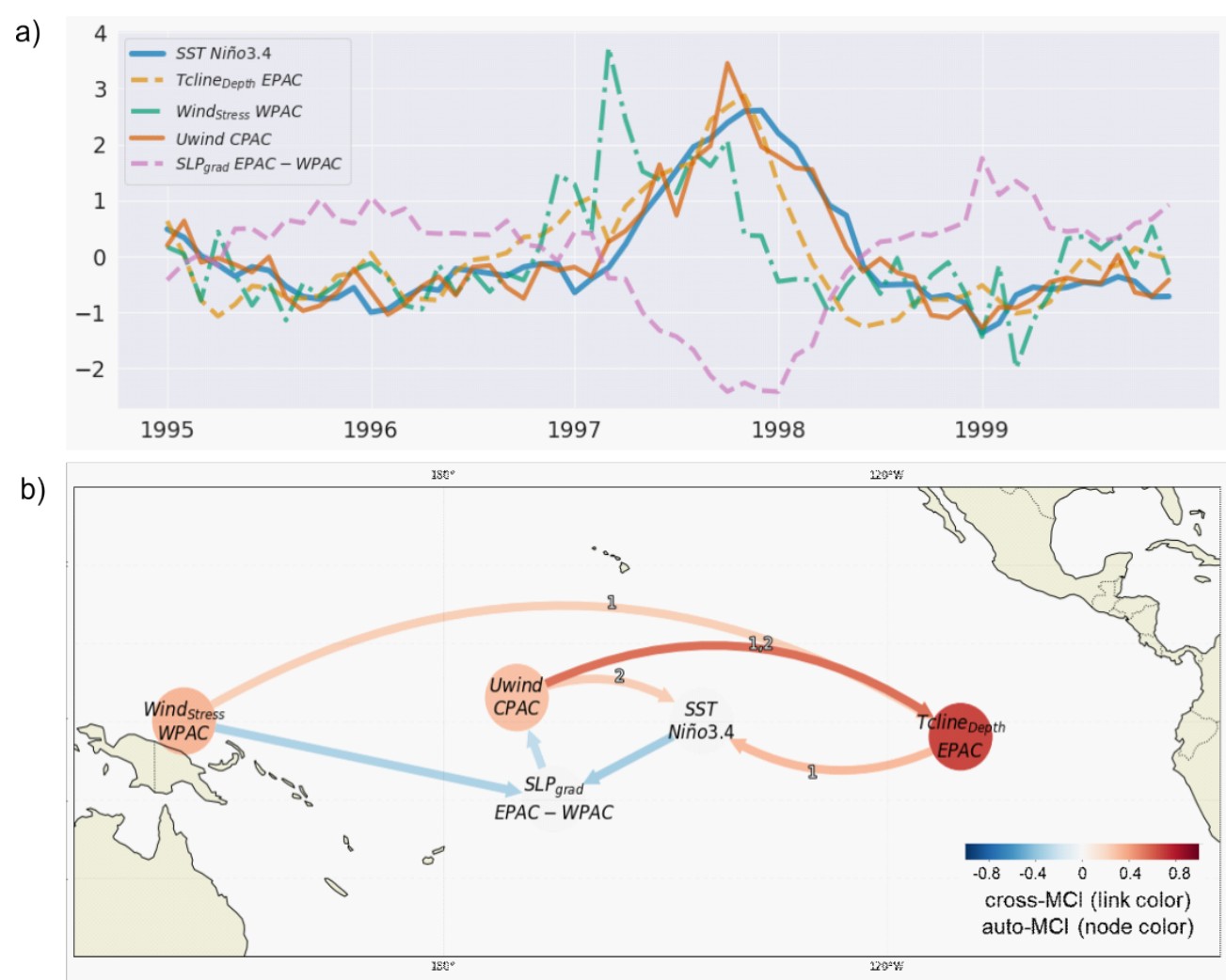

**Figure 2. Causal analysis for the 1997-1998 El Niño event. a) Detrended standardized monthly time series of the variables listed in Table 1 between 1995-1999. b) Causal network representing lagged (curved) and contemporaneous (straight) causal links, constructed by applying PCMCI+ on the time series in (a). Nodes represent the time series associated with each climate variable (see node labels and details in Table 1). Node colors indicate the self-link strengths (autocorrelation) of each time series (auto-MCI, see color bar), and the color of the links denotes the cross-link strengths (cross-MCI, see color bar). The link-associated time lags (unit=1 month) are shown as small labels on the links.**

and Chen, 2021; Lengaigne et al., 2002, 2004). The Bjerknes feedback, which is essential in generating El Niño, is also clearly detected in the resulting causal network. This positive feedback loop involves a coupling between wind, thermocline depth, and SST where a weakening of the equatorial trade winds (westerly wind anomalies) in the West is associated with the deepening of the thermocline in the east. The westerly anomaly winds reduce the upwelling and, along with the deepened thermocline, contribute to warming SSTs in the central and east Pacific. This weakens the East-West SLP and temperature gradients, thus contributing to further weakening of the easterly winds, and so on. These processes are summarized in the causal graph as follows:

- Wind$_{Stress}$ WPAC $\rightarrow$ Tcline$_{Depth}$ EPAC with a 1-month lag and as same-sign response. This link indicates that the westerly wind burst in the West Pacific contributed to an increase in the thermocline depth in the Eastern Pacific during the following month (associated with an eastward propagating downwelling Kelvin wave in the ocean as previously documented). As the thermocline deepens in the east, upwelling brings up warmer water from the thickened thermocline, contributing to the warming of SSTs in the central and eastern equatorial Pacific.

- Wind$_{Stress}$ WPAC $\rightarrow$ SLP$_{grad}$ EPAC-WPAC contemporaneous (zero lag) opposite-sign response. This link indicates that the westerly wind burst in the West Pacific contributes to the decrease in the sea-level pressure gradient between the East and West Pacific. The decreasing SLP gradient means that the sea level pressures over the West Pacific become anomalously higher than those in the East Pacific.

- Tcline$_{Depth}$ EPAC $\rightarrow$ SST Niño3.4 with a 1-month lag and as same-sign response. This link indicates that as the depth of the thermocline in the East Pacific increases, it contributes to the SST anomalies over the Niño3.4 region to increase with a lag of 1 month. This is due to the reduction in the upwelling of colder waters, further contributing to the anomalous warming of SSTs in the central and eastern equatorial Pacific, which represents one of the three components of the Bjerknes feedback.

- SST Niño3.4 $\rightarrow$ SLP$_{grad}$ EPAC-WPAC $\rightarrow$ Uwind CPAC as contemporaneous opposite-sign responses. These links indicate that the rising temperatures over the Niño3.4 region are associated with the decrease in the East-West SLP gradient. This reduction in the SLP gradient contributes to the anomalous westerlies and the weakening of the Pacific Walker circulation as shown by the negative SLP$_{grad}$ EPAC-WPAC $\rightarrow$ Uwind CPAC link. This constitutes the second component of the Bjerknes feedback. As mentioned earlier, the westerly wind burst also contributes to the below-normal SLP gradient at lag zero (Wind$_{Stress}$ WPAC $\rightarrow$ SLP$_{grad}$ EPAC-WPAC).

- Uwind CPAC $\rightarrow$ Tcline$_{Depth}$ EPAC with a 1-month lag and a strong same-sign response. This link shows that the westerly wind anomalies weaken the PWC and strongly contribute to the deepening of the thermocline. While Wind$_{Stress}$ WPAC $\rightarrow$ Tcline$_{Depth}$ EPAC is essential to trigger the 1997/1998 El Niño event, this link is necessary to maintain a strong El Niño state and corresponds to the third component of the Bjerknes feedback, hence, closing the loop. A Uwind CPAC $\rightarrow$ Tcline$_{Depth}$ EPAC link is also found at a 2-month lag with a lower absolute cross-MCI value.

– Uwind CPAC $\rightarrow$ SST Niño3.4 with a 2-month lag and a same-sign response. This lagged link represents the same effect of the Uwind CPAC $\rightarrow$ Tcline$_{Depth}$ EPAC $\rightarrow$ SST Niño3.4 connection at 2-month lag.

To summarize, a causal analysis has been applied to the El Niño event of 1997/1998 to demonstrate the utility of such an analysis in quantifying the connections between physical processes. Previous results are confirmed by this causal analysis in that the westerly wind burst event in the western equatorial Pacific in March triggered an eastward traveling downwelling Kelvin wave in the ocean that deepened the thermocline in the east and shallowed it in the west. As SSTs warmed in the eastern equatorial Pacific, trade winds weakened. This weakening further reduced the upwelling of cold, nutrient-rich waters in the eastern Pacific and instead brought up relatively warmer water from the thickened thermocline, contributing to the warming of SST in the Niño 3.4 region. The identified causal links align with the mechanisms involving the Wind$_{Stress}$ WPAC influencing thermocline depth, which then leads to anomalous warming in the Niño3.4 region, in agreement with extensive previous studies (McPhaden, 1999; Lengaigne et al., 2002, 2003, 2004). Furthermore, the Bjerknes positive feedback is well demonstrated through the weakening of easterly winds, causing a deepening of the thermocline in the east, which consequently warms Niño 3.4 SSTs. The anomalously warm SSTs contribute back to the weakening of easterly winds through adjustment of the East-West SLP gradient and ultimately closing the loop. Although not included in this analysis, anomalous shifts in precipitation patterns are also greatly affected by the SST variations and further contribute to the intensification and maintenance of the 1997/1998 El Niño event.

The advantage of applying causal discovery is that many hypothesized interactions are analyzed simultaneously within one single data frame and multiple connections are estimated while controlling for spurious associations. This represents an advantage over conventional paired regressions, correlations, or multiple regression techniques which cannot always reveal the underlying cause-and-effect relationships among variables.

With this demonstration of the utility of causal analysis in quantifying connections between phenomena that represent previously documented physical processes, we now apply the causal methodology to the less well-understood problem of the nature of the connections between the Pacific and Atlantic.

### 3.2.2 PCMCI+ application to Atlantic-Pacific connections

On the time resolution of the PCMCI+ data frames used in Sect. 4, we note that we use 3-monthly averaged time series of each index (TNA, ATL3, Niño3.4, PNA, NAO, and the PWC$_u$) with four seasons (time steps) per year, as averages of DJF (December, January and February), MAM (March, April and May), JJA (June, July and August) and SON (September, October and November). For the parameter settings of the PCMCI+ algorithm, we set the maximum time lag to 4 time steps ($\tau_{max}$ = 4 [seasons]), meaning that we only investigate teleconnections within a maximum of one-year time lag. Additionally, we estimate contemporaneous links detected within the same season ($\tau_{min}$ = 0). The significance level $\alpha_{pc}$ of all tests carried during the PCMCI+ algorithm in Sect. 4.1 is set to 0.2 to account for the short sample sizes. As more data is available for the pacemaker ensembles (10 simulations each, Sect. 4.2) and the pre-industrial control run (Sect. 4.3), $\alpha_{pc}$ is set to 0.01 and 0.05, respectively. It is important to note that the networks are only causal with respect to the analyzed variables, and more advanced

methods that can deal with hidden variables (Gerhardus and Runge, 2020) may not be suitable for short sample sizes. As we employ multiple simulations corresponding to the same ensemble during the causal analysis of pacemaker time series in Sect 4.2, we utilize the Multidata-PCMCI+ function that allows testing for conditional independencies by combining samples taken from several datasets (i.e. other simulations in the pacemaker ensemble) and learning a single causal graph representing the shared underlying processes. Concerning Sect. 4.3, we show results based on composites selected depending on the phase combination of PDV and AMV (i.e. PDV+/AMV+, PDV-/AMV+, PDV+/AMV- and PDV-/AMV-). A mask is used on the PCMCI+ data frame to select only time steps that satisfy a certain combination. A similar "regime-oriented" analysis is detailed in Fig. 3 of Karmouche et al. (2023). Here, we only use this approach for the pre-industrial control run because larger sample sizes are available and such sampling at lower intervals on the short reanalysis and pacemaker data might produce "spurious" results (Smirnov and Bezruchko, 2012).

Throughout the analysis in Sect. 4, the results are shown for PCMCI+ runs where assumptions have been introduced. Next, we list and discuss all assumptions that were introduced into the analysis. First, to focus on the Atlantic-Pacific interactions, we do not estimate any contemporaneous or lagged dependencies between (1) TNA and ATL3, (2) ATL3 and NAO, (3) ATL3 and PNA, and (4) PNA and $PWC_u$. Although the AZM (ATL3) can be associated with changes in AMM (comprising TNA) through the meridional displacements of the ITCZ, the two modes remain independent and are not considered to affect one another directly (assumption 1, Masson-Delmotte et al., 2023; Murtugudde et al., 2001). The same is true for a direct ATL3-NAO relationship, which was found to be weak and not statistically significant in previous studies (e.g. Wang, 2002, assumption 2). A direct link between PNA and ATL3 is disregarded because there is no proposed physical mechanism for such connection without a major role of ENSO and PWC and also because the main link connecting tropical Atlantic and extratropical Pacific happens through TNA SSTAs' relationship to the pressure system over southeastern United States (assumption 3, Klein et al., 1999; García-Serrano et al., 2017b; Jiang and Li, 2019). Moreover, we consider the PWC to only be connected to PNA through ENSO (e.g. via a poleward-propagating Rossby wave in the case of an El Niño event, Wallace and Gutzler, 1981; Hoskins and Karoly, 1981; Karoly, 1983), hence, we estimate the ENSO-PNA connection only through the Niño3.4-PNA pair (assumption 4). Assumptions 1-4 are held throughout all results shown in Sect. 4. Additionally, during the PCMCI+ analysis of the Pacific pacemaker ensemble (Sect. 4.2.1) we assume that 5) if a contemporaneous connection is estimated between Niño3.4 and any other node, then the link should be oriented from the Niño3.4 node toward the other variable node (assumption 5) because the SSTA is nudged to observed values in the Niño3.4 region. Additionally, to avoid lagged links from $PWC_u$ to Niño3.4 due to their strong relationship, we do not estimate the influences on Niño3.4 from past $PWC_u$ (no $PWC_u \rightarrow$ Niño3.4 link at any lag) in Sect. 4.2.1. Similarly, specific to the PCMCI+ analysis of the Atlantic pacemaker ensemble (Sect. 4.2.2), we assume that 6) if a contemporaneous connection is estimated between an Atlantic SST index (TNA, ATL3) and any other variable, then the link should be oriented from the Atlantic node toward the other node and the same for the direction of the contemporaneous NAO-PNA connection which is assumed as NAO $\rightarrow$ PNA (as this was the most estimated direction when no assumption is introduced, see Supplementary Fig. S13). This is because SSTA over the TNA region is nudged toward observations, and a part of the ATL3 region is included in the linearly tapering buffer zone that extends to the equator (assumption 6). The ensemble-averaged time series (with 25th-75th percentile range shading) for all indices calculated from the Pacific and

Atlantic pacemaker simulations are shown in Supplementary material Figs. S4 and S5, respectively. On another note, 7) the contemporaneous link between Niño3.4 and $PWC_u$ was detected as a conflicting link in several instances when no assumption is introduced (see odd-numbered Figs. S7-15 in Supplementary material). This might be due to the positive Bjerknes feedback loop inadequately captured on the 3-monthly averaged time resolution. Consequently, we assume this connection to be directed as $PWC_u \rightarrow$ Niño3.4 (assumption 7) as this direction occurred the most frequently in the analysis of reanalysis data without any assumptions (excluding unoriented and conflicting links). Assumption 7 is not valid for the PCMCI+ analysis on the Pacific pacemaker ensemble (Sect. 4.2.1), where assumption 5 holds. Finally, to overcome specific instances of conflicting links in Sect. 4.1 between Niño3.4 and ATL3 (analysis of the observed historical period), we 8) assume the orientation of the same-sign contemporaneous Niño3.4-ATL3 connection as Niño3.4 $\rightarrow$ ATL3 (assumption 8). However, it should be noted that the relationship set by assumption 8 is fragile as discussed in Chang et al. (2006) and the same-sign effect proposed by Latif and Grötzner (2000) for ENSO's influence on Atlantic Niños was found to lag by 6 months. All causal graphs obtained without introducing any assumption are shown in odd-numbered supplementary material Figs. S7-15.

## 4   Results

### 4.1   Observed teleconnections in Reanalysis datasets

To investigate the changing interactions and the effect of external forcing during the second half of the 20th and early 21st century from a causal discovery perspective, Fig. 3 demonstrates causal networks of Atlantic-Pacific teleconnections based on indices listed in Sect. 2.4 from the Reanalyses datasets (Sect. 2.1). We first show the long-term state of the Pacific and Atlantic basins represented by the 10-year low-pass filtered PDV and AMV in Fig 3a. Figure 3b shows a sliding window analysis using PCMCI+ where five periods are analyzed (1950-1970, 1960-1980, 1970-1990, 1980-2000, and 1990-2010) using the original observed signal (corresponding to the black curve in Fig. 1). Figure 3c is similar but showing results where indices have been calculated after subtracting MEM, estimating an isolated internal variability (corresponding to the green curve in Fig. 1). Picturing Niño3.4, PNA and $PWC_u$ as variables representing the Pacific, with TNA, ATL3, and NAO representing the Atlantic, we can, for example, estimate how nodes from the Pacific basin are linked to the Atlantic ones, and vice-versa.

As a general note, based on Fig. 3 (all panels), we detect the extensively studied relationship of ENSO and PWC (Trenberth, 1997; Bayr et al., 2014; Zhao and Allen, 2019), illustrated through the strong same-sign contemporaneous causal connection from $PWC_u$ to the Niño3.4 ($PWC_u \rightarrow$ Niño3.4 link). We confirm that positive $PWC_u$ values indicate anomalously weak easterly winds associated with the weakening of PWC and the emergence of El Niño events. With PWC inextricably linked to Niño3.4, a conclusion solidified throughout the results in Sect. 4.2 and 4.3, we consider causal links to and from a $PWC_u$ node to denote a causal relationship associated with ENSO (considering the assumptions listed in Sect. 3.2.2). We also find a weak opposite-sign causal response from $PWC_u$ to Niño3.4 at two season lag (1990-2010 in Fig. 3c) which might be an artifact of the seasonal variations of PWC.

Concerning the Atlantic-Pacific connections, the results suggest a clear decadal change in the interactions. In Fig. 3b, during the 1950-1970 and the 1960-1980 periods the ENSO effect on the tropical Atlantic SST modes is dominant. This is shown

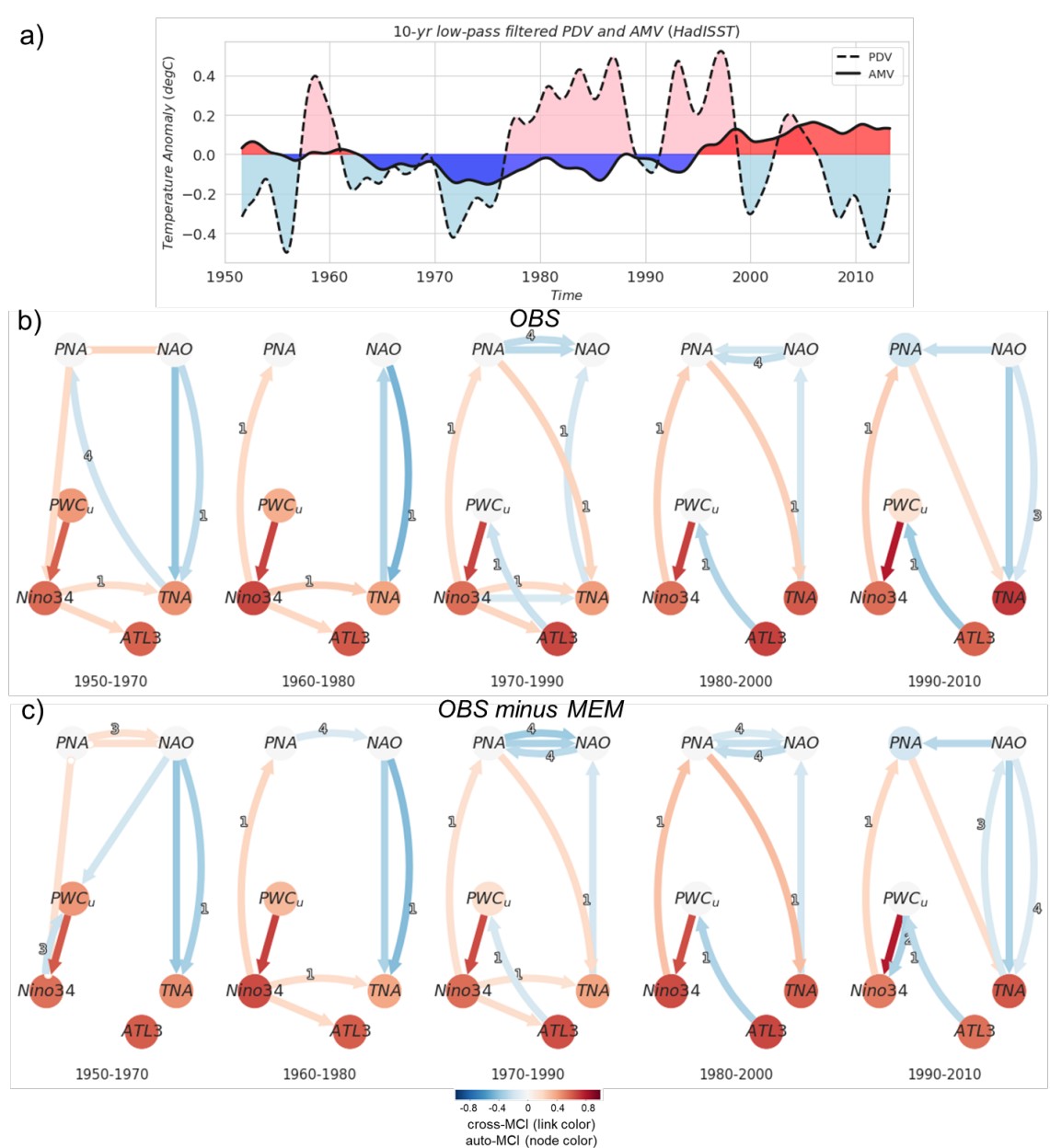

**Figure 3. Observed Atlantic-Pacific interactions. a) 10-year low pass-filtered AMV (solid) and PDV (dashed) from the HadISST dataset, calculated following the definition in Sect. 2.4. b) Sliding window analysis where PCMCI+ is applied for five 20-year windows moving by 10 years (see subtitles for each causal graph). In this panel (b, OBS), the algorithm is run on the original time series before removing MEM. The link-associated time lags (unit=1 season i.e 3 months) are shown as small labels on the curved links. c) Similar to (b) but using data where the MEM is removed (OBS minus MEM).**

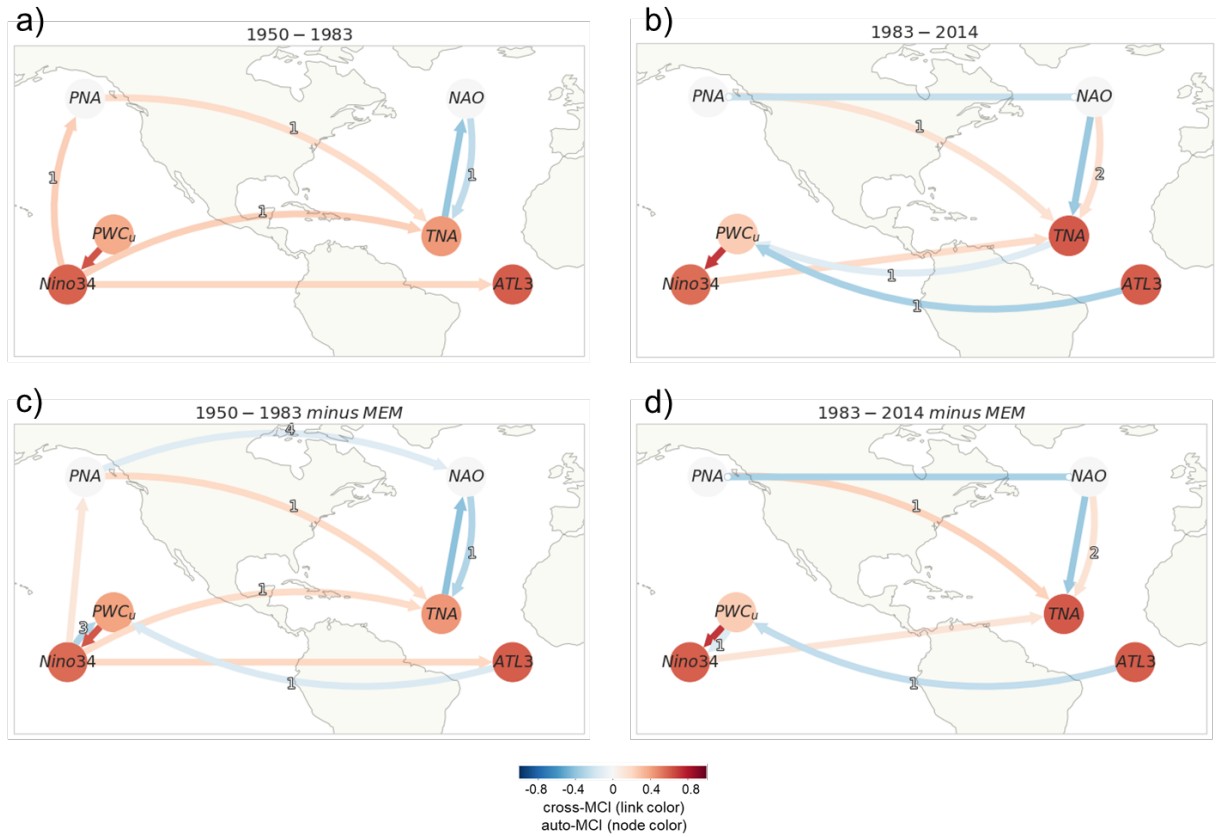

**Figure 4. Causal networks representing Atlantic-Pacific teleconnections for 1950-1983 (left column) vs 1983-2014 (right column) in the Reanalyses datasets. (a) Constructed by applying PCMCI+ the 1950-1983 period on the original time series before removing MEM. (b) Same as (a) but for the 1983-2014 period. (c) same as (a), but with indices calculated after removing MEM. (d) Same as (c) but for the 1983-2014 period.**

as a same-sign (positive cross-MCI) Niño3.4 → ATL3 link at lag zero and Niño3.4 → TNA at lag 1. The first two periods correspond to mainly negative PDV and AMV phases (see panel a). During the subsequent 1970-1990 period, where PDV switches to a positive phase after the mid-1970s and AMV is in a cold phase, the results show the emergence of a negative-sign response from ATL3 to PWC$_u$. We also notice during this period a weakening of the Niño3.4 links to TNA and ATL3, but an extra-tropical link to TNA is established through PNA (lagged PNA → TNA during 1970-1990). During the 1980-2000 and 1990-2010 periods, there is a decay of the same-sign influence from ENSO to the tropical Atlantic as no direct Niño3.4 → TNA nor Niño3.4 → ATL3 links were detected (Fig. 3b and c). Meanwhile, a strengthening of the ATL3 to PWC link is apparent during the last two periods as AMV trends toward and maintains a warm state. This suggests that as the Pacific and the Atlantic SSTAs fluctuate on the (multi)decadal timescale, changes in the interactions between modes of interannual variability are inextricable. These changes are not only confined to the tropics. In the extra-tropical region, a change in the connection

between PNA and NAO is also detected. While this is detected as a same-sign contemporaneous unoriented adjacency (PNA o-o NAO) during the first 20-year window 1950-1970 (and not detected during the 1960-1980 period), this connection was estimated as contemporaneous negative during the three subsequent periods, as suggested by previous literature. The PNA-NAO connection was also found with a 4-season lag, which might be detected as a false positive due to the high correlation to the previous winter. In Fig. 3c, where the causal networks take into account the effect of subtracting MEM, we detect a similar pattern to Fig. 3b with small differences. Generally, a switch after the 1970-1990 period from a Pacific-dominated regime to an Atlantic-dominated one is maintained. Apart from the non-estimated Niño3.4 links to TNA and ATL3 during the first period, the most noteworthy differences after removing the externally forced signal are: i) a slightly stronger link for the extra-tropical route connecting PNA to TNA at 1-season lag during 1980-2000 in comparison to 1970-1990, and ii) no pronounced increase in the strength of the 1-season lagged $ATL3 \rightarrow PWC_u$ link between the 1980-2000 and 1990-2010 windows (compared to the same window periods when MEM was not subtracted, Fig. 3c vs b), suggesting contributions of external forcings. A decadal regime shift in the Pacific-Atlantic interactions falls in agreement with findings from previous studies (e.g. Meehl et al., 2020; Park et al., 2023b). We further analyze longer periods similar to the ones proposed in (Park et al., 2023b), namely the Pacific-driven period (1950-1983) and the Atlantic-driven period (1983-2014). The causal networks obtained through PCMCI+ are shown in Fig. 4 before (panels a and b) and after (panels c and d) removing the externally forced signal.

**Pacific-driven period 1950-1983**

The Pacific-driven 1950-1983 is found to be dominated by a same-sign effect from ENSO on both TNA (1-season lagged) and ATL3 (lag zero). The effect on TNA is detected through both tropical and extra-tropical routes. During an El Niño event, the weakened Walker circulation allows an eastward shift in the maximum convection center from the Maritime Continent to the central equatorial Pacific. This tropical convection triggers a poleward-propagating Rossby wave, which extends into the midlatitudes, constituting the PNA pattern (Wallace and Gutzler, 1981; Hoskins and Karoly, 1981; Karoly, 1983). This teleconnection linking the equatorial and extra-tropical Pacific is detected as a lagged same-sign causal link from Niño3.4 to PNA during 1950-1983 (Fig. 4a and c). The wave pattern associated with PNA contributes to the formation of an anomalous low-pressure center over the southeast United States and the Caribbean. The presence of the PNA pattern results in anomalous southwesterly winds over the TNA region. The negative rainfall anomaly over the western Pacific and the Atlantic region, caused by the reversed Walker circulation during El Niño, plays a role in inducing this anomalous low-pressure center over the southeast United States. Additionally, the suppressed heating response in the Atlantic region, resembling the Gill-like pattern (Matsuno, 1966; Gill, 1980) might explain the Niño3.4 $\rightarrow$ ATL3 contemporaneous positive link (Fig. 4a and c) and also contributes to the development of anticyclonic circulation and southwesterly wind anomalies over the TNA region (García-Serrano et al., 2017b; Jiang and Li, 2019). The combined effect of these extratropical and tropical routes leads to southwesterly wind anomalies that weaken the northeasterly trade winds, reduce evaporation, and induce SSTA warming over the TNA region (Trenberth, 1997; Wallace and Gutzler, 1981; García-Serrano et al., 2017b; Jiang and Li, 2019; Meehl et al., 2020; Casselman et al., 2021; Park et al., 2023b). The two routes for the ENSO effect on TNA, which is predominant during the 1950-1983 period, can be seen through the causal networks in Fig. 4a and c showing two same-sign (lagged) links from the Pacific to TNA: PNA→TNA (1 season) and Niño3.4→TNA (1 season).

The Pacific-driven regime's causal graph when MEM is subtracted (Fig. 4c) shows a 4-season lagged PNA→NAO link, incon-
sistent with previous studies on the linkage between North Atlantic and North Pacific modes of atmospheric circulation that
suggest the contemporaneous negative link between PNA and NAO. This 4-season lagged link, as mentioned earlier could be
the artifact a spurious correlation to the next year's winter. Honda et al. (2001) conducted a study on the period between 1979
and 1994 and discovered a negative correlation (-0.7) between the intensities of the Aleutian and Icelandic lows (low-pressure
centers of the PNA and NAO, respectively). Song et al. (2009) concluded that the strongest negative correlations between
PNA and NAO occur with no time lag and within a range of 10-day lags. By analyzing reanalysis datasets, Pinto et al. (2010)
found no significant anti-correlation between PNA and NAO between 1950 and the mid-1970s, but this was detected during
the sub-period 1973–1994. A period of weak PNA-NAO coupling might explain the undetected contemporaneous PNA→NAO
negative links during the Pacific-driven period (Fig. 4a) and the 1960-1980 window in Fig. 3b. According to Soulard and Lin
(2016), it is the absence of tropical forcing from ENSO that strengthens the relationship between PNA and NAO. On the other
hand, observational uncertainty before the satellite era can also be a reason that lagged and/or positive links were detected
between NAO and PNA (1950-1970 in Fig. 3b and c) instead of contemporaneous negative links. On the Atlantic side, apart
from the influence of ENSO and PNA on TNA, NAO is also found to be connected to TNA SSTAs (contemporaneous nega-
tive TNA→NAO and 1-season lagged TNA→NAO links in Fig. 4a and c). Proposed mechanisms for this connection involve
changes in pressure gradients between the Azores high and the Icelandic low which alter trade winds, heat fluxes, and SSTs.
Reduced northeasterly trade winds contribute to trapping warm SSTAs over the TNA region as less latent heat is released
into the atmosphere (Cassou and Terray, 2001; Lee et al., 2008). Although a direct link connecting either Niño3.4 (or $PWC_u$)
to NAO has only been detected during the 1950-1970 window (Fig. 3c), the NAO is thought to affect the interplay between
ENSO and TNA which is further complicated by the fact that ENSO can also influence the NAO through extratropical pathways
(García-Serrano et al., 2017a; Casselman et al., 2021). If we consider the causal links that connect the SST modes (Niño3.4
to TNA and/or ATL3) directly or through $PWC_u$ and PNA, then these results support the hypothesis that it was the Pacific
SSTs mainly driving the same-sign response on the Atlantic SSTs the first half of the analyzed period 1950-2014 (Meehl et al.,
2020; Park et al., 2023b). Most importantly, comparing Fig. 4a to c reveals the emergence of a weak negative lagged ATL3 →
$PWC_u$ link when external forcing is removed during the first period. Although negligible, MEM estimates a net cooling effect
of external forcing on equatorial (and north tropical) SSTs during the 1950-1983 period (see signal in red for TNA and ATL3
in Fig. 1b and f, respectively).

**Atlantic-driven period 1983-2014**

    Unlike the first period, equatorial and north tropical Atlantic warming is observed during the second period (see TNA and
ATL3 in Fig. 1b and f, and AMV in Fig. 3a), bringing anomalous westerly wind and enhanced precipitation (Park et al.,
2023a). The Rossby wave energy associated with the increased precipitation propagates toward the tropical Pacific. Combined
with the modulated Walker circulation, this induces easterly wind anomalies over the equatorial Pacific, favoring the devel-
opment of La Niña events (Ham et al., 2013b; Park et al., 2022, 2023b). Overall, during 1983-2014 the Atlantic Niño and
TNA are hypothesized to have similar fingerprints concerning the effect on ENSO (Park et al., 2021). The Atlantic effect on
ENSO is illustrated in Fig. 4b and d through the 1-season lagged ATL3 →$PWC_u$ strongly negative link, an effect that reaches

the Niño3.4 node (strong PWC$_u$→Niño3.4 link). The externally forced causal graph in Fig. 4b also features a weak 1-season lagged TNA→PWC$_u$ which vanishes when MEM is removed (Fig. 4d). While the atmospheric bridge connecting the equatorial and the extra-tropical Pacific was detected as Niño3.4→PNA links in Fig. 4a-c, this link was not detected during the second period (Fig. 4b and d). The NAO is found to drive changes in TNA (Fig. 4b and d), in contrast with the direction estimated during the Pacific-driven regime (TNA→NAO, Fig. 4a and c). The PNA effect on TNA is detected as a weak link in Fig. 4b and stronger when external forcing was removed (Fig. 4d). Contrary to the first period, the contemporaneous negative PNA connection to NAO (Honda et al., 2001; Song et al., 2009; Pinto et al., 2010) is detected during the second period as a contemporaneous unoriented adjacency (PNA o-o NAO in Fig. 4b and d). Song et al. (2009) explain the anti-correlation between day-to-day variability of the Aleutian low and Icelandic low as the result of the anomalous Rossby wave-breaking events associated with the PNA pattern. The authors show that when the PNA is in a positive (negative) phase, there is more (less) Rossby wave breaking over the North Pacific (Atlantic), which can weaken or split (strengthen) the polar vortex over that region. This can then affect the jet stream and the storm tracks over the North Atlantic, leading to a negative (positive) NAO phase (Song et al., 2009). Lagged negative causal PNA→NAO and NAO→PNA links were also detected during the two windows spanning 1970-2000 in Fig. 3b and c.

It appears from the causal networks from the 1950-1983 period that the effect of MEM was to suppress a weak negative sign effect from ATL3 on ENSO through PWC (Fig. 4c vs 4a). Whereas for the following 1983-2014 period, external forcing seems to favor a weak contribution of TNA to the predominant negative sign effect of Atlantic on ENSO. The non-stationary relationship of ENSO to the tropical Atlantic from one regime to another is influenced by the decadal changes in the background mean state. Special emphasis needs to be given to the effect of AMV in the interplay between the Atlantic Niño and the tropical Pacific. The interconnection between AZM and ENSO is favored during the negative phase of the AMV when a shallower thermocline enhances equatorial SST variability (Martín-Rey et al., 2014b; Park and Li, 2018). Furthermore, Wang et al. (2017b) propose an Atlantic capacitor mechanism, wherein ENSO impacts TNA through an atmospheric bridge (termed as "charging"). Subsequently, the Atlantic influences the following ENSO by "discharging" via a subtropical teleconnection. Specifically, when AMV is trending to its negative phase, the impact of ENSO on TNA becomes amplified and has a more prolonged effect (Park and Li, 2018). This was the case during the first two analyzed windows in Fig. 3 and the 1950-1983 period in Fig. 4 when AMV was trending towards its negative phase. During the following period, AMV was trending back from a negative to a positive phase and the opposite was observed (decaying strength). The anomalously warm North Atlantic SSTAs during the positive AMV phase favor the strengthening of the PWC, which ultimately brings upwelled cold water to consequently cool down the central equatorial Pacific. The specific contributions of internal variability to these regime changes remain unclear. Zhang et al. (2019) emphasizes the influence of the thermohaline circulation, particularly the Atlantic Meridional Overturning Circulation (AMOC), on the multidecadal changes in Atlantic SSTs. Their study also suggests that the interannual fluctuations are primarily driven by wind-induced changes in turbulent heat fluxes. On the other hand, a series of recent papers (Booth et al., 2012; Mann et al., 2014; Bellucci et al., 2017; Watanabe and Tatebe, 2019; Klavans et al., 2022) show growing evidence of an increasing effect of external forcing on the AMV and its lead/lag association with the AMOC.

This implies that internal variability and external radiative forcing contribute to the decadal SST variations over the Atlantic (Meehl et al., 2016; Park et al., 2019; Meehl et al., 2020; Klavans et al., 2022; Park et al., 2023b).

## 4.2 Pacemaker simulations

### 4.2.1 Pacific pacemaker

To scrutinize the potential causal dependencies between the modes, we use a 10-member ensemble of the CESM2 Pacific
pacemaker simulations (see Sect. 2.2) where tropical SSTAs have been nudged towards observed values (maintaining ENSO evolution, see Supplementary Fig. S4c). The rest of the coupled model is free to evolve, resulting in different climate variations outside the nudging region depending on the single realizations' initial conditions. The range of possible outcomes for Atlantic SSTAs is then governed by contributions from internal variability, CMIP6 time-varying external forcing, and the potential cross-basin contributions from the Pacific according to the model's dynamics. The variations of the TNA and ATL3 indices in
the Pacific pacemaker simulations before and after removing the CMIP6 time-evolving external forcing (MEM) are shown in Supplementary Fig. S4a and f, respectively. To illustrate the long-term state of the Pacific and Atlantic basins in the Pacific pacemaker ensemble, we show in Fig. 5a and b, the low-pass filtered PDV and AMV time series averaged across all ensemble members (shadings denote the 25th to 75th percentile range) where orange (blue) curves denote the ensemble time series before (after) subtracting MEM. The observed time series of both low-pass filtered indices before (after) subtracting MEM are
shown as black (green) dashed lines. The PDV is defined north of 20°N and is outside the nudging region, and is the same for AMV in the Atlantic. This means that the SST in the PDV and AMV regions (see Sect 2.4) are free to evolve according to the CESM2 model coupling, CMIP6 external forcing, and the tapering applied. There are several discrepancies before the 1990s between the pacemaker-simulated and the observed indices (before and after subtracting MEM) for both PDV and AMV. However, Fig 5a shows that the 25th-75th percentile range of pacemaker-simulated PDV reduces significantly after the 1990s
when external forcing is enhanced (orange shading in Fig. 5a). Removing external forcing brought PDV closer to the observed range of anomalies but with a wider range of values, i.e. restoring internal variability. The pacemaker-simulated AMV follows closely the observed range of anomalies before and after MEM is removed, especially after the 1990s. This further shows the increasing effects of external forcing in steering North Atlantic warming during the most recent decades of the analyzed period and the modulation of variability over the Pacific.

Similar to Fig. 3b and c, in Fig. 5c and d, we show a similar analysis applied to the Pacific pacemaker ensemble using the Multidata-PCMCI+ function. For every window, the causal graph represents the average causal dependencies from all 10 ensemble members. As more samples are available for the Pacific pacemaker, we lower the significance threshold $\alpha_{pc}$ to 0.01 to show only robust (low uncertainty) links. We also note that the causal graphs shown in this section are subject to special assumptions (due to the nudging) as mentioned at the end of section 3.2.2. In most of the periods from the five analyzed
windows in Fig. 5c, we see the same-sign effect of the Pacific on the Atlantic either through the tropical (Niño3.4→TNA) pathway or the extra-tropical one (PNA→TNA) or both. This is also true after subtracting MEM (Fig. 5d).

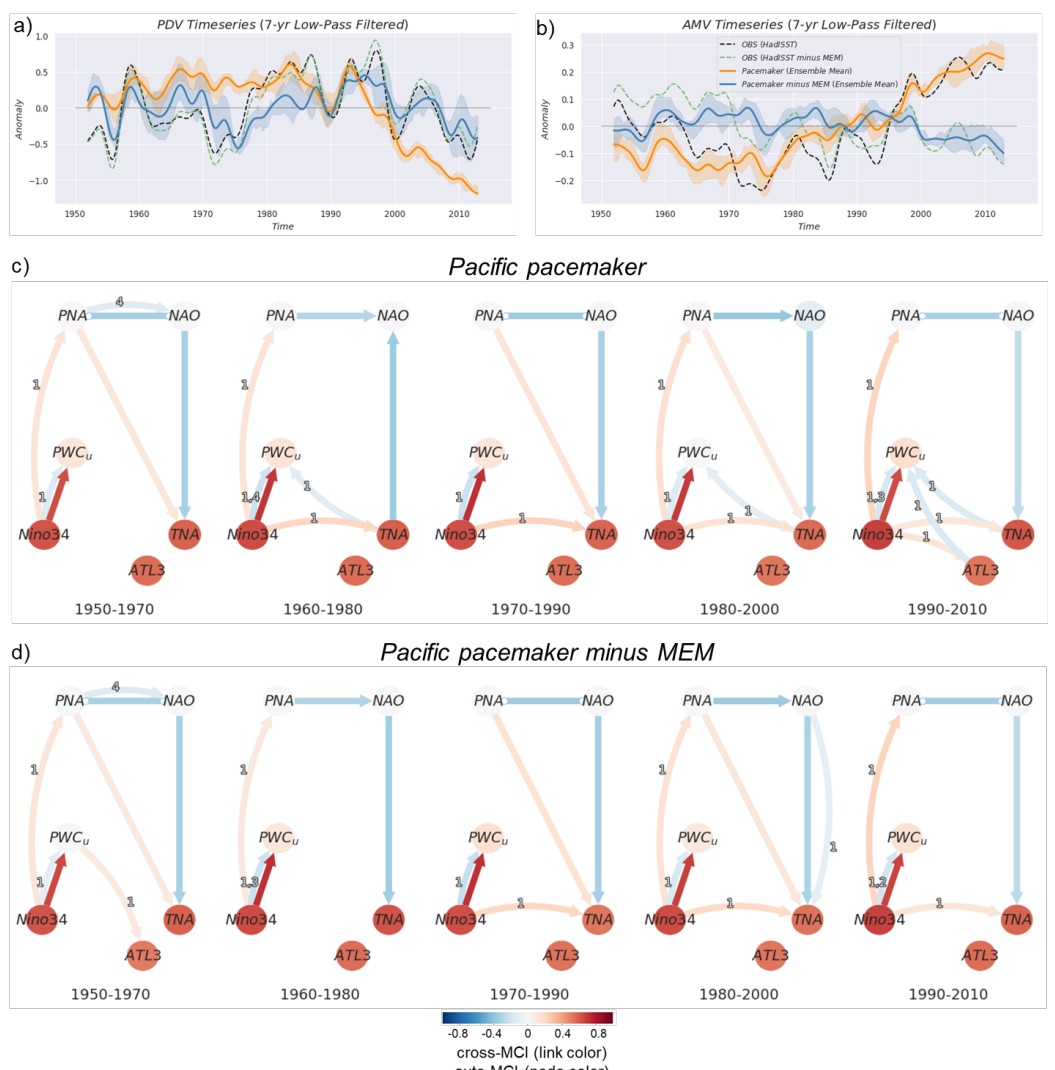

**Figure 5. Pacific pacemaker simulations where tropical Pacific SSTAs have been nudged toward observations (see Sect. 2.2). a) Pacific pacemaker ensemble average of the 7-year (Lanczos) low-pass filtered PDV time series (10 members) together with observations (HadISST, see legend in next panel) following definition in Sect. 2.4 for the 1950-2014 period. Solid lines denote the pacemaker ensemble-averaged time series before (orange) and after (blue) subtracting MEM. Shadings show the ensemble's 25th-75th percentile range. Dashed lines are for the same index calculated from the HadISST dataset before (black) and after subtracting MEM (green). b) Same as (a) but showing the AMV index. c) Same as Fig. 3b but using the Multidata-PCMCI+ function. Here, every window (see subtitle under each graph) shows a single causal graph representing the dependencies estimated from combined samples taken from all 10 pacemaker simulations. d) Same as (c) but for indices calculated after MEM is removed from every single pacemaker simulation.**

The prominent feature of the causal analysis on the Pacific pacemaker simulations before removing external forcing (Fig. 5c) is the ability to distinguish a period with an Atlantic effect on PWC-related wind anomalies. All periods after 1950-1970 show the equatorial Pacific SSTAs driving a same-sign effect on the TNA SSTAs (and ATL3 during the last window) and triggering a negative-sign response from the Atlantic onto the Pacific during the most recent periods when external forcing is strongest (via TNA→ $PWC_u$). The two last windows in the panel c also show a decay of ENSO's same-sign effect accompanied with detection of lagged negative links from tropical Atlantic SSTA to PWC. Unlike observations, removing external forcing in Fig. 5d results in the complete disappearance of TNA→ $PWC_u$. This means that the forced Atlantic response has been strongly modulating the local Pacific response to external forcing during the 1990s and early 21st century through remote impacts on PWC. Literature suggests that ENSO (Maher et al., 2015, 2018) and its decadal imprint, PDV (Allen et al., 2014; Dong et al., 2014), have contributions from external drivers, especially volcanic and anthropogenic aerosols. Other studies show that recent period of global warming hiatus is the result of anthropogenic aerosols modulating the phase of PDV rather than canceling out other warming effects (Kaufmann et al., 2011; Smith et al., 2016).

The simulations from the pacemaker ensemble are in agreement with the proposed contemporaneous negative PNA relationship to NAO (Honda et al., 2001). This was mostly detected as contemporaneous PNA→NAO or PNA o-o NAO links. While this section provides insight onto the effects of ENSO and external forcing in the interaction between the two basins, the next section provides the results of a similar analysis using an ensemble of Atlantic pacemaker simulations to investigate the combined effects of the Atlantic and external forcing on the Pacific.

### 4.2.2 Atlantic pacemaker

Similar to section 4.2.1, Fig. 6 shows the PDV and AMV time series (panel a and b) and the Multidata-PCMCI+ results from the Atlantic pacemaker ensemble (panel c and d). While there are notable differences between the Atlantic pacemaker-simulated and observed PDV, these differences are reduced significantly after the 1990s. In the North Atlantic, small differences between the observed and Atlantic pacemaker-simulated AMV are mostly due to observational uncertainty (HadISST vs ERSSTv3b). As mentioned earlier in this section, we assume links only originate from the Atlantic affecting the Pacific (north Atlantic SSTAs nudged toward observations). Similar to observations (Fig. 3) and the Pacific pacemaker ensemble (Fig. 5), the results in Fig. 6 demonstrate that the same-sign response between TNA and Niño3.4 is detected during the first three windows spanning from 1950 to 1990 and decays afterward. Additionally, the negative sign effect of the tropical Atlantic on PWC (and ultimately ENSO) is prevalent throughout all analyzed periods (except the first window in Fig. 6c), with little difference before and after subtracting MEM. This suggests that internal variability alone can generate the decadal change in the interactions between the Pacific and the Atlantic. Nevertheless, there are small contributions that appear to be the result of external forcing and are illustrated through slightly stronger opposite-sign links during 1970-1990 (ATL3 → $PWCu$) and 1990-2010 (TNA → $PWCu$) when the externally forced signal is kept (Fig. 6c). An interesting feature is that as the PDV switches to a negative phase after the 1990s, the negative sign impact of the Atlantic on PWC becomes mainly detected from TNA and not ATL3 (Fig. 6c,d). In the Atlantic pacemaker ensemble, this feature seems to be independent of external forcing as there are considerable differences in the magnitude of the AMV index before and after subtracting MEM (orange vs blue time series in Fig. 6b), but

no pronounced discrepancies between causal graphs of Fig. 6c vs d. We note that the directions shown for the contemporaneous connections with TNA are based on assumption 8 discussed in Sect. 3.2.2 (i.e. based on the nudging). Causal graphs obtained when no assumption is introduced during the analysis of the Atlantic pacemaker ensemble are shown in Supplementary Fig. S13.

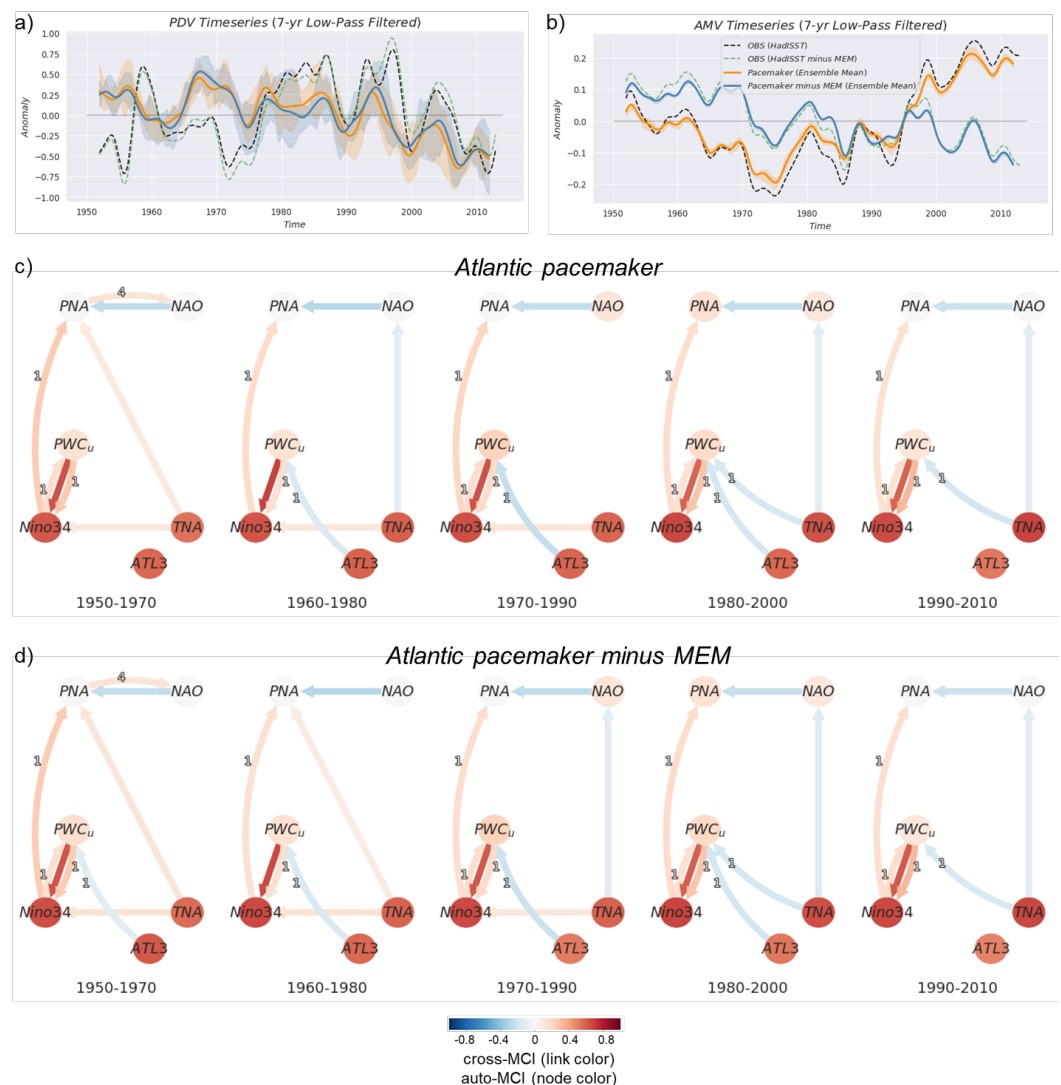

**Figure 6. Atlantic pacemaker simulations where North Atlantic SSTAs have been nudged toward observations (see Sect. 2.2). Panels a-d here are similar to Fig. 5a-d but for the Atlantic pacemaker ensemble.**

An important consideration is the difference in the externally forced signal between the Atlantic pacemaker simulation and MEM. Unlike the Pacific pacemaker ensemble and MEM, the externally forced signal in the Atlantic pacemaker simulations is based on the CESM1 model, which includes CMIP5 (and not CMIP6) forcing. The same estimate of external forcing

(MEM based on CMIP6 large ensemble models) is removed from both pacemaker ensembles. This presents a limitation to the assessment of external forcing on the CESM1 model in this section. On the other hand, consistency in the estimation of MEM as the forced signal at least addresses the issue of major differences due to volcanic aerosol forcing in the second half of the 19th century and early 21st century. These differences have been reported and mainly attributed to small-to-moderate eruptions included in CMIP6 natural forcing but not in CMIP5 simulations (Fyfe et al., 2021). The different sources of external forcing as well as the overestimation of most recent warming trends, complicate the precise attribution of external contributions to the Atlantic-Pacific interactions during the last decades of the historical record. To further test whether the observed teleconnections would arise only from internal climate variability (and natural external forcing), the next section presents results from a pre-industrial control run of the CESM2 model.

## 4.3 Pre-industrial control simulations

In the absence of anthropogenic forcing, Atlantic-Pacific interactions inherently depend on the mean state of the two basins. To illustrate this, we use 250 years (3-monthly averaged, 1000 times steps) from the CESM2 pre-industrial run (CESM2 piControl). In Fig. 7 a-d, the PCMCI+ causal graphs are shown for specific phase combinations depending on the sign of the low pass filtered (13-year Lanczos) time series of PDV and AMV shown in Fig. 7e.

First, most of the observed teleconnections (and pacemaker-simulated) are also detected during the pre-industrial control run. The same-sign effect of the Pacific on the Atlantic is estimated during three out of the four regimes. During PDV+/AMV+, PDV-/AMV+, PDV+/AMV- (Fig. 7a-c), it is detected as both tropical pathway from Niño3.4 (and PWCu) to TNA and/or ATL3 (1-2 season lag), and extratropical route from PNA to TNA (0-1 season lag). On the other hand, the negative sign effect of the Atlantic on the Pacific is detected from ATL3 to $PWC_u$ only when PDV is in a negative phase (PDV-/AMV+, PDV-/AMV-, Fig. 7b and d).

Moreover, the results also show that the two basins are "better" interconnected when PDV and AMV are out of phase (PDV-/AMV+, PDV+/AMV-; Fig 7b and c) as there are more links with higher cross-MCI values compared to when the two indices (AMV and PDV) are in phase i.e. PDV+/AMV+ and PDV-/AMV- (Fig. 7a and d). Specifically, the same-sign effect of ENSO on tropical Atlantic SSTAs is more pronounced when the decadal North Pacific SSTA is in a warm state while the North Atlantic is in a cold state (PDV+/AMV-, Fig. 7c). Conversely, when these phases are inverted (PDV-/AMV+), it is the extra tropical atmospheric variability modes that are strongly connected (PNA o-o NAO, Fig. 7b). The absence of strong El Niño forcing during that regime might explain the strength of the PNA-NAO in Fig. 7b, as suggested by Lin and Derome (2004); Soulard and Lin (2016).

In contrast to the reanalysis and Pacific pacemaker ensemble, the causal graphs from the CESM2 piControl run show only instances of NAO contributing to TNA and not the other way around, suggesting stronger dependence of tropical North Atlantic SST on atmospheric variability forcing given the 250 years of unforced simulation. Another discrepancy is that the in-phase combination of AMV and PDV (PDV+/AMV+ or PDV-/AMV-) seems to be favored and long-lasting during the pre-industrial era in comparison to observations. During the finite sample of CESM2 piControl used here, AMV and PDV were in phase 58% of the time, but in observations (ERSSTv5) and the CESM2 historical simulations (11 ensemble members) only 43% and

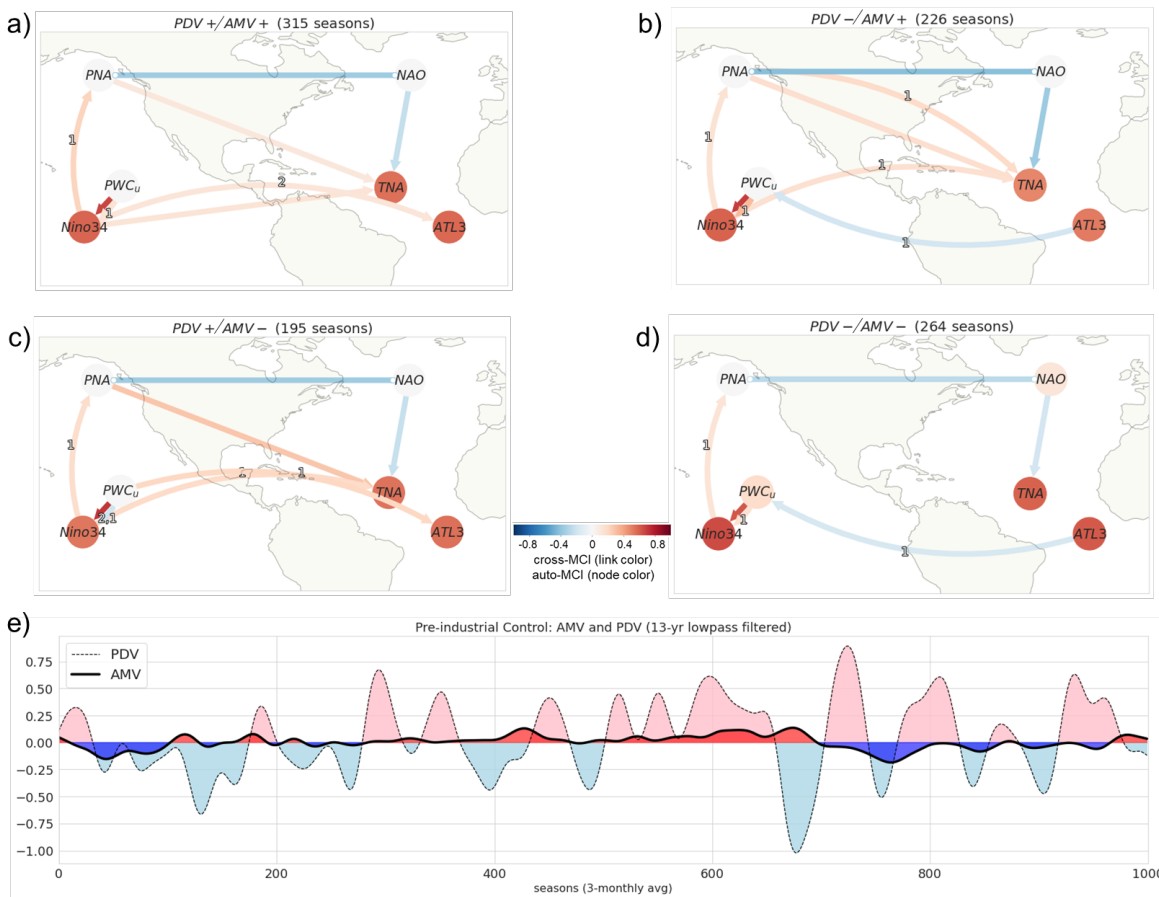

**Figure 7. Regime-oriented PCMCI+ analysis of the CESM2 pre-industrial control run. a) Causal graph where only time steps corresponding to either a) PDV+/AMV+, b) PDV-/AMV+, c) PDV+/AMV-, d) PDV-/AMV- regime are considered. The regimes are defined according to the phase of e) smoothed AMV and PDV indices (13-year low-pass filtered) illustrating the decadal internal variability over the Atlantic and Pacific for 250 years (1000 time steps [seasons (3-monthly averages)]. Positive (negative) phases are shaded in pink (light blue) and red (blue) for PDV and AMV, respectively.)**

20% respectively, between 1900 and 2014 according to (Karmouche et al., 2023). Does the external forcing signal observed during the historical period drive an "unstable" state where contrasting responses between the two basins are favored due to the prevalence of an out-of-phase regime? To answer this question, longer periods and more CMIP6 large ensemble models with

historical simulations and their pre-industrial counterparts should be analyzed.

From the analysis of the pre-industrial run, the following could be summarized: (i) "Pacific-driven" and "Atlantic-driven" periods happen naturally in the absence of anthropogenic forcing. (ii) This does not reject the possibility that human-induced forcing contributed greatly to at least the observed changes after the 1990s. (iii) The strength, sign, and direction of the cross-basins interactions are modulated by the long-term state of both basins (PDV and AMV).

## 5  Discussion and Conclusions

Serving as a "proof of concept" within the domain of comprehending complex climate phenomena, the El Niño event of 1997/1998 presents a notable example showcasing the efficacy of causal discovery in discerning the underlying drivers that govern well-established connections among contributing coupled processes (methods, Sect. 3.2.1). Hereupon, the scope extends beyond ENSO, delving into the less-explored topic of Atlantic-Pacific interactions.

Based on causal graphs derived from reanalysis data and Pacific pacemaker simulations before and after separating the externally forced signal (Figs. 3-5), we find the Pacific's same-sign influence on the Atlantic during the first half of the analyzed period, aligning with prior studies (Meehl et al., 2020; Park et al., 2023b). Estimated positive links from Niño3.4 (or $PWC_u$) to TNA (or ATL3) represent the tropical pathway where El Niño and the modified Walker circulation prompt equatorial Atlantic anticyclonic activity, weakening trade winds, and warming tropical North Atlantic as a result of reduced evaporation. Other pathways recognized through PNA and NAO have also been detected. Positive causal links Niño3.4→PNA and PNA→TNA highlight the extra-tropical path for the Pacific-induced effect on TNA. El Niño-associated Rossby wave propagation enhances Southeast United States low-pressure center (positive PNA phase), weakening North Atlantic trade winds and promoting TNA warming (Klein et al., 1999; García-Serrano et al., 2017b; Jiang and Li, 2019). The transition from a Pacific-driven to an Atlantic-driven regime is evident in reanalysis and pacemaker simulations through the weakening (or decay) of the same-sign response between ENSO and tropical Atlantic in the second half of the analyzed period (e.g. Fig. 3 c and d), accompanied by the emergence and strengthening of ATL3's and TNA's negative sign effects on $PWC_u$ (Fig. 3 c and d, 5c). Based on reanalysis data, the 1983-2014 causal graphs (Fig. 4b and d) reveal lagged negative ATL3→$PWC_u$ and TNA→$PWC_u$ links, denoting intensifying easterly trade winds affecting ENSO. Proposed mechanisms involve modified Walker circulation in the equatorial Atlantic for the effect of ATL3 (AZM) on PWC and the Rossby wave energy generated by enhanced precipitation for the effect of anomalously warm TNA on the subtropical Pacific, which causes easterly wind anomalies and La Niña-like cooling (Wang et al., 2017b; Park and Li, 2018; Park et al., 2022). Observations show a negative contemporaneous NAO-PNA relationship only starting the 1970-1990 window (and the 1983-2014 period in Fig. 4). Earlier research proposed a robust negative correlation between Aleutian and Icelandic lows only during specific historical sub-periods (e.g., mid-1970s to mid-1990s; Honda et al., 2001; Pinto et al., 2010). This might clarify the absence of contemporaneous negative PNA-NAO links during the causal analysis for the Pacific-driven periods in reanalysis (Fig. 4a and c). Conversely, the simultaneous negative PNA-NAO link was consistently identified across both pacemaker ensembles and the pre-industrial control runs. This link could be the result of Rossby wave-breaking events connecting PNA to opposite NAO phases (Song et al., 2009) or simply because, particularly in certain seasons, the two modes are spatially overlapping projections of the same variability pattern, connecting the Aleutian and Icelandic lows (Soulard and Lin, 2016).

A new aspect of the nature of Atlantic-Pacific connections is shown by separating external forcing from internal variability by subtracting CMIP6 MEM in the causal analysis, yielding insights into external forcing's impact on Pacific-Atlantic interactions. Although MEM removal had modest effects on the Atlantic pacemaker causal graphs (Fig. 6b vs c), pronounced impacts emerged during the analysis of the Pacific pacemaker ensemble (Fig. 5b vs c), especially during the most recent pe-

riods. The 1980-2000 and the 1990-2010 periods in the Pacific pacemaker results show the emergence of ATL3→$PWC_u$ and TNA→$PWC_u$ only when the external forcing is not separated, underscoring external forcing's primary role in the post-1980s periods. In contrast, the negative "ATL3 to ENSO" effect (via PWC) persisted through the second half of the analyzed period despite MEM removal in the reanalysis run (Fig. 3b vs c and Fig. 4b vs d). In general, the Pacific pacemaker results showed that the Pacific signal was probably externally forced but not through local processes. A major role played by external forcing in the negative effect of TNA on ENSO is consistent with recent studies highlighting an increasing impact of external forcing on North Atlantic SSTAs and associated effects in recent decades (Murphy et al., 2017; Klavans et al., 2022; He et al., 2023). This brings additional insights into the debate over the attribution of the recently observed strengthening of the PWC. Our results show that external forcing contributions modulated by the Atlantic might have amplified the recent PWC strengthening, in contrast with previous studies suggesting a dominant role of internal variability (Chung et al., 2019).

The analysis of the historical record (from 1950 to early 21$^{st}$ century) suggests external forcing's potential contributions, yet does not exclude the role of internal variability driven by the Pacific and Atlantic long-term states (PDV and AMV). To explore Atlantic-Pacific internal variability interactions in unforced conditions, we analyzed 250 years from the CESM2 pre-industrial control run, following a regime-oriented approach. The causal graphs underscore the Pacific's impact on tropical Pacific variability through the tropical and extra-tropical pathways during three out of four regimes (all except PDV-/AMV-). The interconnection between the Atlantic and Pacific basins was found strongest when the PDV and AMV are out of phase (PDV+/AMV-, PDV-/AMV+), with more links and higher cross-MCI values compared to when they are in phase. Additionally, the negative sign effect ATL3→$PWC_u$ was only detected when PDV was negative. The pre-industrial control analysis indicates that both contrasting response regimes arise naturally without anthropogenic external forcing, influenced by the long-term states of the Pacific and Atlantic basins.

The results in Figs. 5-7 suggest that the coupling between Atlantic and Pacific SSTs is moderately strong in the CESM models, explaining the appearance of links connecting Niño3.4 (and/or PWCu) to TNA (and/or ATL3) in most causal networks shown in Sect 4.2 and Sect. 4.3. The coupling is also realistic as it captures the observed patterns of interdependencies shown in Fig. 3. The main difference between the reanalysis and Pacific pacemaker causal networks remains during the most recent decades after removing external forcing. Differences might be due to the overestimation of the 1998-2013 global warming rate in CMIP6 climate models (McBride et al., 2021; Smith et al., 2021; Fyfe et al., 2021; Smith and Forster, 2021; Tokarska et al., 2020a, b), and inherently in MEM. This additionally complicates the separation of external forcing in the CESM1 Atlantic Pacemaker ensemble as it includes CMIP5 forcing. Different factors contributing to such overestimation in CMIP6 have been proposed, including the high equilibrium climate sensitivity (ECS) that results in too strong warming responding to anthropogenic GHGs or too weak cooling responding to aerosols (IPCC, 2013; Tokarska et al., 2020a; Schlund et al., 2020; Smith and Forster, 2021; Wei et al., 2021). According to Wei et al. (2021), ECS only plays a partial role in the failure of most CMIP6 models in simulating the early 2000s global warming slowdown. Instead, the authors attribute the discrepancy between observed and CMIP6-simulated warming trends mostly to the models' deficiencies in simulating major modes of internal variability at interannual, interdecadal, and multidecadal scales, thus excluding their potential effects (e.g. the cooling effect of PDV switching to a negative phase in the early 2000s).

In a concluding remark, the authors would like to highlight that causal discovery is a powerful tool to assess the physical mechanisms of Atlantic-Pacific interactions. However, a careful selection of the potential variables representing the analyzed mechanisms and the length of the time series are crucial for a robust application of causal analysis and reliable interpretation of detected connections. Therefore in order to make credible conclusions based on the application of causal discovery, it is important to accurately determine the causal assumptions, clarify the correct confounding variables, and analyze the physically consistent interactions at their most relevant time scales to achieve robust results.

We also recognize the potential benefits of a more granular analysis, such as splitting each time series into four quadrants (representing four seasons) to analyze the effects on a particular season (e.g. on the peak season of a certain variable or summer vs winter periods). However, implementing such a quadrant-based method would dramatically shorten the data frame and impair the causal analysis, especially for reanalysis and pacemaker ensembles (where short periods are analyzed). Despite this limitation, the adopted methodology which considers the complete-year time series (3-monthly averaged) identifies the overall effect considering all time steps (seasons) and provides valuable insights into the seasonal lag and strength of causal relationships. Naturally, the most significant links (under the $\alpha_{pc}$ threshold) are the ones that come out on the causal graph, and which should always be interpreted according to background knowledge (e.g. with respect to the peak season of the variables analyzed). Given the complexity of causal graphs, the existence of prior knowledge about underlying physical processes can significantly improve the analysis and interpretation of the results. Therefore, the integration of expert assumptions helps avoid conflicting adjacencies to further estimate the strength of the connections.

Finally, this study aims to enhance our understanding of the teleconnections between the Atlantic and Pacific oceans and their variability under different regimes. The findings emphasize the significance of external forcing, particularly in the most recent regime, and highlight the roles of tropical and extra-tropical pathways, and internal variability in shaping SST variability over the Pacific and Atlantic basins on different timescales. Further research is warranted to refine our knowledge of these complex interactions and improve model simulations to capture the observed teleconnections more accurately. External forcing represented by the CMIP6 MEM has contributions from natural (e.g. solar radiation, volcanic eruptions) and anthropogenic sources (e.g. aerosols, GHGs) with time and space-varying effects. Therefore, we encourage further analysis using simulations with single external forcing sources (e.g. aerosol-only or GHG-only simulations) to increase the understanding and attribution of the observed changes in the climate system.

*Code and data availability.* The CESM2 Pacific pacemaker ensemble dataset can be found here: https://www.earthsystemgrid.org/dataset/ ucar.cgd.cesm2.pacific.pacemaker.html[dataset] (last access: 29 April 2024). The CESM1 Atlantic pacemaker ensemble dataset can be found here: https://www.earthsystemgrid.org/dataset/ucar.cgd.ccsm4.ATL-PACEMAKER.html (last access: 29 April 2024). The complete description and documentation of the Pacific and Atlantic pacemaker datasets are available on the Climate Variability and Change Working Group's (CVCWG) webpage (https://www.cesm.ucar.edu/working-groups/climate/simulations/, last access: 29 April 2024). The Earth System Model Evaluation Tool (ESMValTool Righi et al., 2020) has been used for preprocessing and calculating the CMIP6 MEM. The Tigramite package for causal discovery is available under the following public GitHub repository: https://github.com/jakobrunge/tigramite [code], last access: 29 April 2024, Runge et al. (2023). Details on the Multidata-PCMCI functionality can also be found on Tigramite's GitHub repository under:

https://github.com/jakobrunge/tigramite/blob/master/tutorials/dataset_challenges/tigramite_tutorial_multiple_datasets.ipynb (last access: 29 April 2024). The code to produce the figures and supplementary material is accessible at the time of publication of the paper in the following

GitHub repository: https://github.com/EyringMLClimateGroup/karmouche24esd_AtlanticPacificPacemaker_Causality [code].

*Author contributions.* SK lead the study, the writing of the manuscript, and performed all the analysis. All co-authors contributed to the concept of the study, to the interpretation of the results, and to the writing of the manuscript

*Competing interests.* The authors declare no competing interests.

*Acknowledgements.* We would like to thank the editor and two anonymous reviewers for their valuable suggestions and comments. This study

was funded by the European Research Council (ERC) Synergy Grant "Understanding and modeling the Earth System with Machine Learning (USMILE)" under the Horizon 2020 research and innovation programme (Grant agreement No. 855187), the European Union's Horizon 2020 research and innovation programme under Grant Agreement 101003536 (ESM2025—Earth System Models for the Future) as well as through project "European Eddy-Rich ESMs" (EERIE) from the European Union's Horizon Europe research and innovation programme under Grant Agreement No. 101081383. All UK Partners in EERIE are funded by UK Research and Innovation (UKRI) under the UK

government's Horizon Europe funding guarantee (grant numbers 10057890, 10049639, 10040510, 10040984). ETH Zürich's contribution to EERIE is funded by the Swiss State Secretariat for Education, research and Innovation (SERI) under contract 22.00366. EG is supported by Central Research Development Fund at the University of Bremen, Funding No: ZF04A/2023/FB1/Galytska Evgenia. We acknowledge the World Climate Research Programme's (WCRP's) Working Group on Coupled Modelling (WGCM), and we thank the CMIP-participating climate-modeling groups for producing and making available their model output. We also thank the CVCWG for publicly publishing their

CESM pacemaker simulations. This work used resources of the Deutsches Klimarechenzentrum (DKRZ) granted by its Scientific Steering Committee (WLA) under project no. bd1083. We also acknowledge the use of AI tools for tasks including formatting, spell-checking, and the automated generation of references in the Bibtex format.

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
