# Peer review of "Changing effects of external forcing on Atlantic-Pacific interactions"

_EGUsphere, 2023_

## Author Comment (AC1)

**Reply to RC1**

Newly added figures and their captions:

[Figure]

**Figure 2. Causal analysis of the for the 1997-1998 El Niño event. a)** Detrended standardized monthly time series of the variables listed in Table 1 between 1995-1999. **b)** Causal network representing lagged (curved) and contemporaneous (straight) causal links, constructed by applying PCMCI+ on the time series in (a). Nodes represent the time series associated with each climate variable (see node labels and details in Table 1). Node colors indicate the self-link strengths of each time series (auto-MCI, see color bar), and the color of the links denotes the cross-link strengths (cross-MCI, see color bar). The link-associated time lags (unit=1 month) are shown as small labels on the links.

[Figure]

Figure 3. Observed Atlantic-Pacific interactions. a) 10-year low pass-filtered AMV (solid) and PDV (dashed) from the HadISST dataset, calculated following the definition in Sect. 2.4. b) Sliding window analysis where PCMCI+ is applied for five 20-year windows moving by 10 years (see subtitles for each causal graph). In this panel (b, OBS), the algorithm is run on the original time series before removing MEM. The link-associated time lags (unit=1 season i.e 3 months) are shown as small labels on the curved links. c) Similar to (b) but using data where the MEM is removed (OBS minus MEM).

[Figure]

**Figure 4. Causal networks representing Atlantic-Pacific teleconnections for 1950-1983 (left column) vs 1983-2014 (right column) in the Reanalyses datasets. (a) Constructed by applying PCMCI+ the 1950-1983 period on the original time series before removing MEM. (b) Same as (a) but for the 1983-2014 period. (c) same as (a), but with indices calculated after removing MEM. (d) Same as (c) but for the 1983-2014 period.**

[Figure]

[Figure]

**Figure 6. Atlantic pacemaker simulations where North Atlantic SSTAs have been nudged toward observations (see Sect. 2.2). Panels a-d here are similar to Fig. 5a-d but for the Atlantic pacemaker ensemble.**

[Figure]

**Figure 7. Regime-oriented PCMCI+ analysis of the CESM2 pre-industrial control run. a) Causal graph where only time steps corresponding to either a) PDV+/AMV+, b) PDV-/AMV+, c) PDV+/AMV-, d) PDV-/AMV- regime are considered. The regimes are defined according to the phase of e) smoothed AMV and PDV indices (13-year low-pass filtered) illustrating the decadal internal variability over the Atlantic and Pacific for 250 years (1000 time steps [seasons (3-monthly averages)]. Positive (negative) phases are shaded in pink (light blue) and red (blue) for PDV and AMV, respectively.)**

**Changing effects of external forcing on Atlantic-Pacific interactions**

Soufiane Karmouche[1,2], Evgenia Galytska[1,2], Gerald A. Meehl[3], Jakob Runge[4,5], Katja Weigel[1,2], and Veronika Eyring[2,1]

[1]University of Bremen, Institute of Environmental Physics (IUP), Bremen, Germany
[2]Deutsches Zentrum für Luft- und Raumfahrt e.V. (DLR), Institut für Physik der Atmosphäre, Oberpfaffenhofen, Germany
[3]Climate and Global Dynamics Laboratory, National Center for Atmospheric Research (NCAR), Boulder, CO, USA
[4]Deutsches Zentrum für Luft- und Raumfahrt e.V. (DLR), Institut für Datenwissenschaften, Jena, Germany
[5]Fachgebiet Klimainformatik, Technische Universität Berlin, Berlin, Germany

**Correspondence:** Soufiane Karmouche (sou_kar@uni-bremen.de)

**List of Figures**

**1 Supplementary Material for Section 3.2.1 (Proof of Concept)**

Figures S1 and S2 show two of the pre-processing steps taken to decide the conditional independence test (Fig. S1) and the maximum time lag $\tau_{max}$ (Fig. S2) during the PCMCI+ analysis. The non-Gaussian marginal distribution of the time series and the use of *RobustParcorr* is only applicable for Sect. 3.2.1. The density plots for the data used in Sect. 4 (not shown) did not reveal such distributions, hence, *Parcorr* was used as conditional independence test there. Similar to Fig. S2, the lag function plots for the variables in Sect. 4 (not shown) are analyzed to decide $\tau_{max}$. Except for a PNA-NAO connection at 8-season lag (having no physical basis), all dependencies decay after a maximum lag of 4 time lag steps, hence the use of $\tau_{max} = 4$ seasons.

Figure S3 shows the p-values (uncertainty) associated with the PCMCI+ estimated coefficients during the proof-of-concept analysis shown in Fig. 2.

[Figure]

**Figure S1. Density plots derived from kernel density estimation (analogous to histograms) are shown off-diagonal. Along the diagonal, the marginal distributions are non-Gaussian for most variables. As mentioned in Sect 3.2, the skewed distributions are handled with the *RobustParcorr* conditional independence test in the proof of concept (Sect. 3.2.1).**

**2   Supplementary Material for Section 4**

In this supplementary section, we show the averaged time series of the six variables used in the analysis of Atlantic-Pacific interactions (Sect. 4.2) for the Pacific pacemaker ensemble (Fig. S4) and the Atlantic pacemaker ensemble (Fig. S5). The p-value matrices for the PCMCI+ estimated dependencies (the coefficients in all causal graphs) in Sect. 4. These present an uncertainty measure of the respective cross(auto)-MCI coefficients. The links shown on the causal graph have p-values lower than $\alpha_{pc}$ and are shown in red in the odd-numbered Figs S7-15. The significant threshold $\alpha_{pc}$ is set to 0.2 in Sect. 4.1, 0.01 in Sect. 4.2,

[Figure]

**Figure S2. Lag function plot showing the lagged dependencies between the variables in Sect. 3.2.1 for six months. Most dependencies decay by $\tau = 3$ time steps, hence the choice for $\tau_{max} = 3$ [months] in the proof of concept (Sect. 3.2.1).**

60    and 0.05 in Sect. 4.3. We also show in the even-numbered Figs S6-14, the causal graphs obtained through PCMCI+ when no background knowledge is introduced. As discussed in methods (Sect. 3), the PCMCI+ networks might contain conflicting links $(X_i^t \times - \times X_j^t)$ at lag zero. The potential differences between the links shown here (Supplementary material) and the ones on the main manuscript lie in the adjacencies that were not considered by PCMCI+ due to the assumptions introduced and listed in Sect. 3.2.2 (based on background knowledge, sensitive analyses, and inspection of the conflicting edges).

[Figure]

**Figure S3. Lagged p-value matrix for the cross-MCI coefficients in the causal graph shown Fig. 2b.** The x-axis of each scatter subplot shows the time lag from $\tau_{min} = 0$ to $\tau_{max} = 3$ [months]. The p-values are shown on the y-axis for each pair (see variable names left and top of each subplot) and denote the uncertainty of each estimated dependency. The p-values below the significance threshold $\alpha_{pc}$ (here $\alpha_{pc}$=0.1) are shown in red.

[Figure]

**Figure S4. Ensemble-averaged indices from the Pacific pacemaker ensemble (10 members) for a) TNA, b) PNA, c) Niño3.4, d) PWCu, e) NAO, and f) ATL3 for the 1950-2014 period. The time series in orange (blue) represent the indices calculated before (after) subtracting MEM. Shadings denote the 25th-75th percentile range.**

[Figure]

**Figure S5. Same as Fig. S4 but for the Atlantic pacemaker ensemble (10 members) for the 1950-2013 period.**

[Figure]

**Figure S6. Lagged p-value matrices for the coefficients on the causal graphs shown in Fig. 3b and c. The Figure is rotated 90° to the left to fit the page format. a) The are five subplots for the five windows analyzed in each panel (see corresponding Fig 3b) each resembling the p-values matrix shown in Fig. S3 but for the indices and parameters used in Sect. 4.1. Here, $\tau_{min} = 0$ and $\tau_{max} = 4$ [seasons i.e. 3-monthly averages]. The p-values below the significance threshold $\alpha_{pc}$ (here $\alpha_{pc}$=0.2) are shown in red. Panel (a) corresponds to causal graphs in Fig. 3b (OBS, see the title and corresponding subtitles in Fig. 3b for each window period). The pairs of adjacencies that were not estimated following assumptions (listed in Sect. 3.2.2) have p-values approaching 1. b) Same as (a) but correspond to graphs shown in Fig. 3c (OBS minus MEM).**

[Figure]

**Figure S7. Causal graphs with no background knowledge. a)** PCMCI+ causal networks corresponding to the ones shown in Fig. 3b obtained when no assumptions are introduced i.e. all dependencies between all variables at all lags (up to $\tau_{max}$) are considered. These might contain conflicting ($X_i^t \times - \times X_j^t$). **b)** Same as (a) but for the causal graphs of Fig. 3c.

[Figure]

**Figure S8.** Similar to Figs S3 and S6 but for causal graphs shown Fig 4 (panels a-d here correspond to panels a-d in Fig. 4). Significance threshold $\alpha_{pc}$ is set to 0.2 here.

[Figure]

**Figure S9. Similar to Fig. S7 but for the causal graphs in Fig. 4 (panels a-d here correspond to panels a-d in Fig. 4).**

[Figure]

**Figure S10. Similar to Figs S6 but for causal graphs shown Fig. 5b and c. The significance threshold $\alpha_{pc}$ is set to 0.01 here.**

[Figure]

**Figure S11. Similar to Fig. S7 but for the causal graphs in Fig. 5b and c.**

[Figure]

**Figure S12. Similar to Figs S6 but for causal graphs shown Fig. 6b and c. The significance threshold $\alpha_{pc}$ is set to 0.01 here.**

[Figure]

**Figure S13. Similar to Fig. S7 but for the causal graphs in Fig. 6b and c.**

[Figure]

**Figure S14.** Similar to Fig. S6 but for the causal graphs shown in Fig. 7a-d. The significance threshold $\alpha_{pc}$ is set to 0.05 here.

[Figure]

**Figure S15. Similar to Fig. S7 but for corresponding to the causal graphs in Fig. 8 (panels a-d here correspond to panels a-d in Fig. 8)**

---

## Author Response (AR1)

**Reply to RC1**

We thank the referee for his/her constructive comments. We addressed every aspect in the comments and made a major revision to our analysis and manuscript. We enumerate the Referees' Comments in blue in "quotation marks" and we address them below in **bold**. We use the following notation: P1 L1 means Page 1, Line 1 to refer to text from the old manuscript. We put in ***italic and bold*** the text added to the new version of the manuscript. We note that this revision takes into account major second referee comments (see reply to RC2). The newly added Figures and their captions (and supplementary material) can be found at the very end of this document (and attached).

1. "While the connections between ENSO and TNA are properly considered, I am puzzled why another, very prominent Atlantic-Pacific connection is missing: The one related to the Atlantic Nino (measure e.g. by the ATL3 index). Numerous papers have been addressing how ENSO influences the Atlantic Nino, and how the Atlantic Nino influences ENSO, and the decadal variations of these connections have also been discussed. In a manuscript with the title Atlantic-Pacific interactions, this connection can certainly not be ignored. I strongly encourage the authors to include the ATL3 index, and perhaps other indices that potentially could act as links."

**We followed the Referee's recommendation and included the Atlantic Zonal Mode (AZM, represented by the ATL3 index) in our analysis. Our revised manuscript shows the new results with a sixth added variable (Section 4, Results) for all runs (observations, pacemaker simulations, and pre-industrial control run). Being the primary mode of tropical Atlantic variability, we believe including ATL3 made the analysis more robust in terms of capturing the effect of the Atlantic on the Pacific. For the causal analysis of the reanalysis data, the new results include a sliding window analysis using PCMCI+ and show that the opposite-sign effect from ATL3 to ENSO (Nino3.4 or PWCu) became stronger and dominant from the 1980s window onward (new Fig. 3). This increase in strength is attributed to the phase switch of AMV and is intensified with external forcing. The Pacific pacemaker results show that the negative effect of the Atlantic (either from TNA or ATL3) on PWC is only detected as a result of external forcing during the 1980-2000 and 1990-2010 windows (new Fig. 5). The Atlantic pacemaker results (now included following major comment #4 in RC2; new Fig. 6) show that the Atlantic effect is dominated by the AZM and TNA only overtakes in the presence of strong external forcing. In the results of the pre-industrial control (now showing an AMV/PDV phase-dependent analysis, see main comment #2 in RC2), the Atlantic effect on the Pacific is found only during the negative Phase of the PDV, and both basins are strongly interconnected when PDV and AMV are out of phase.**

2. "Proof of concept and all other connections involving ENSO: While most of the connections regarding the ENSO mechanism itself appear plausible, I am puzzled by the fact the the positive Bjerknes feedback does not properly come out. Clearly Nino3.4 should impact the central Pacific winds, which is one of the main ingredients of the Bjerknes feedback. This is probably at zero lag, therefore a straight link? By the way, I do not see any straight links in the analysis/figures."

**We agree with the Referee's comment that the previous analysis did not estimate lag zero links and did not include any index for a zonal wind in the Central Pacific. In the new version of the manuscript, we replaced the precipitation anomalies**

(Precip central CPAC) with the eastward wind anomalies in the central Pacific (Uwind CPAC). We also changed the minimum time lag to 0 (to estimate contemporaneous links in addition to the lagged ones) to better capture the Bjerknes feedback. We believe that the new Fig. 2 now clearly shows the proficiency of the PCMCI+ in detecting different aspects related to El Nino 1997-1998 event. We still find that the wind stress in the west Pacific (following the March 1997 westerly wind burst) contributes to the: 1) reducing the sea level pressure gradient between the east and the west Pacific with no time lag and 2) deepening of the thermocline in the east Pacific. The wind stress WPAC is the only nodein the new graph which only has the outgoing links. Apart from the clear detection of the role of the westerly wind burst during the 1997-1998 El Nino, we also detect the Bjerknes feedback which is essential to generating and maintaining an El Nino state. This is seen in the causal graph as a feedback loop consisting of the three main components:

1) link from the thermocline depth to the SSTs in the Nino3.4 region (1-month lag). 2) a contemporaneous connection showing: an increase of Nino3.4 SST ☐ anomalous decline of East-West Pacific SLP gradient ☐ increase in westerly winds anomalies. 3) The weakened easterly winds are found to be strongly linked to the deepening of the thermocline in the east (1-month lag) to close the positive feedback loop.

3.     " Links of anything more than 1 season need special attention in their explanation. Surely, no purely atmospheric bridge could explain lags of more than 1 season (probably not even that). How can PNA and NAO impact each other at 7 month lag? There most be complex deep physical analysis here, which is completely lacking. Perhaps these links are indirect, though some SST patterns, but we can only speculate this right now. It seems that the technique used cannot answer such kind of questions. "

We thank the Referee for this valuable comment. We agree, that 7-months PNA and NAO lag is indeed challenging to interpret. To give credit to the method and preserve a better detection power, we decided to change the maximum time lag parameter in the analysis of Sect. 4 to tau_max = 4 seasons. This not only resolves the physically unexplainable lagged links (between PNA and NAO) but is important for the robustness of the results following the addition of a sixth variable and small sample sizes. Such links could be indeed spurious due to the combination of short samples at large maximum lags and/or due to unobserved variables.

4.     "Pacemaker experiments: If the SSTs in the Pacific are prescribed, how can the causal Network come to the conclusion that PWC is forcing ENSO? So, this technique is not able to detect trivial directions? Clearly the only logical explanation is that Nino3.4 is forcing PWC, but the Network does not detect this (apart from just in one case).

To give credit to the method, the issue of the earlier pacemaker results stems from the post processing of the PCMCI+ graphs. In the old manuscript, we transformed the original cross-MCI values (link strengths) to standardized linear link coefficients estimated after fitting a linear mediation model to the set of parents estimated by PCMCI+. The idea behind this was to better quantify the causal links by providing a measure of the change in units of standard deviations. The challenge that rises with this approach is that it requires a fully oriented Directed Acyclic Graph (DAG), which is not always the case for the resulting PCMCI+ graphs, as they might contain unoriented links at lag-zero, either as contemporaneous unortiented adjacencies

(denoted as 'o-o') or conflicting links (represented by 'x-x'). Transforming such links (exclusive to lag-zero) was based in most cases on a majority ruling among the CESM2 pacemaker ensemble (as previously explained in P8 L201-207). We recognize this was not an ideal approach especially when it comes to providing the ensemble average graphs. The method originally detected several unoriented links between Nino3.4 and PWC in the individual pacemaker simulation runs (compared to both observations and pre-industrial control) but those undirected links were not taken into account for deciding the overall direction in the transformation to fully oriented DAG. To address these issues we revised our analysis by: 1) showing the original cross-MCI values (instead of the standardized link coefficients) for expressing link strengths. This is in order to tolerate links that denote contemporaneous adjacencies ('o-o'). In the remaining cases of conflicting links ('x-x'), we 2) introduce background knowledge as assumptions based on sensitivity tests and/or well-established physical processes. These assumptions are used to address the main issue presented by this referee comment. The Pacific pacemaker ensemble causal analysis is now supported now by a background assumption that states: in the case of a detected lag-zero adjacency between the nudged variable (i.e. Nino3.4) and another variable, the link should be directed from the nudged variable to the other variable. We add detailed explanations of these new adjustments into our methodology section in P8 L193 (Section 3.2):

*The Tigramite package offers the ability to plot results in the form of a causal graph where nodes represent the time series associated with each variable. In these graphs, the node color shows the auto-MCI value (auto-correlation i.e. self-links) and the link color indicates the cross-MCI value (i.e. link strength) with blue indicating opposite-sign (negative) inter-dependency and red indicating same-sign (positive) inter-dependency. The link-associated time lags are shown as small labels on the curved links. If a link is detected at different lags, the indicated lags are sorted by link strength (i.e. by the absolute cross-MCI value). Contemporaneous links (at lag zero) are represented by straight lines. In the context of causal links between variables $X_i$ and $X_j$ at time t, the possible link types considered for any time lag (τ) are non-adjacent links (i.e., the pair is not directly connected) and direct links from $X_i$ at time t−τ to $X_j$ at time t($X_{t-τ} → X_t$)). For τ = 0 additional possible link types are opposite direct links ($X_t ← X_t$, unoriented links ($X_t$ o−o $X_t$), and conflicting links ($X_t$ ×−× $X_t$) which can occur due to finite sample effects or violations of assumptions. While we tolerate the presence of unoriented adjacencies due to Markov equivalence ($X_t$ o−o $X_t$, hereafter denoted by "o-o" symbol) in the causal graphs of the upcoming sections, we introduce assumptions for the cases where conflicting links ($X_t$ x−x $X_t$) occur. For the analysis of the Pacific and Atlantic pacemaker ensembles (Sect 4.2), we introduce these assumptions on the basis of which variables have been nudged towards observed values. For example, in the PCMCI+ tests on the Pacific pacemaker simulations, the implemented assumption states that if the method detects a contemporaneous adjacency between Nino3.4 and any other variable X, then the link should be directed from Nino3.4 to X (Nino3.4 → X). This is because Nino3.4 is the pacemaker as the SSTA is nudged towards observations in that region. We follow the same approach for the Atlantic pacemaker, where the assumptions presume only outgoing contemporaneous links from TNA (as the SSTAs in that region are nudged to observations). Some other assumptions are introduced to overcome specific cases of "x−x"-type links by presuming their orientation according to background knowledge (well-defined physics from previous studies) and/or sensitivity tests (using different conditional independence tests, different $α_{pc}$ and/or sliding window analysis). While no*

*assumption is predefined in next section's proof of concept, the assumptions introduced in each of the Results subsections (Sects. 4.1-4.3) are explained in detail at the end of Sect. 3.2.2. In Supplementary material, we show all original graphs which possibly contain conflicting edges ($X_i$ x−x $X_j$) and where all dependencies between all variables at all lags are considered (i.e. no assumptions introduced)*

Concerning the ensemble averaging of the individual pacemaker graphs, we now utilize the Multidata-PCMCI+ function which allows independence tests to be drawn on a pool of samples combining multiple datasets and produce a single graph that summarizes the shared underlying processes. Combined with the sliding window analysis (instead of two fixed periods), we also believe the Multidata function addresses robustness questions raised in main comment 2 in RC2. Here, we add to our manuscript in Section 3.2.2 an introduction to Multidata-PCMCI+ (used in Sect. 4.2) and the regime-oriented approach (followed in Sect. 4.3). We add the following in P12 L301:

*As we employ multiple simulations corresponding to the same ensemble during the causal analysis of pacemaker time series in Sect 4.2, we utilize the Multidata-PCMCI+ function that allows testing for conditional independencies by combining samples taken from several datasets (i.e. other simulations in the pacemaker ensemble) and learning a single causal graph representing the shared underlying processes. Concerning Sect. 4.3, we show results based on composites selected depending on the phase combination of PDV and AMV (i.e. PDV+/AMV+, PDV−/AMV+, PDV+/AMV− and PDV-/AMV-). A mask is used on the PCMCI+ data frame to select only time steps that satisfy a certain combination. A similar "regime-oriented" analysis is detailed in Fig. 3 of (Karmouche et al., 2023). Here, we only use this approach for the pre-industrial control run because larger sample sizes are available and such sampling at lower intervals on the short reanalysis and pacemaker data might produce "spurious" results (Smirnov and Bezruchko, 2012).*

We also add the description concerning the assumptions (after the addition above) in P12 L301:

*Throughout the analysis in Sect. 4, the results are shown for PCMCI+ runs where assumptions have been introduced. Next, we list and discuss all assumptions that were introduced into the analysis. First, to focus on the Atlantic-Pacific interactions, we do not estimate any contemporaneous or lagged dependencies between (1) TNA and ATL3, (2) ATL3 and NAO, (3) ATL3 and PNA, and (4) PNA and PWCu. Although the AZM (ATL3) can be associated with changes in AMM (comprising TNA) through the meridional displacements of the ITCZ, the two modes remain independent and are not considered to affect one another directly (assumption 1, Cassou et al., 2021; Murtugudde et al., 2001). The same is true for a direct ATL3-NAO relationship, which was found to be weak and not statistically significant in previous studies (e.g. Wang, 2002, assumption 2). A direct link between PNA and ATL3 is disregarded because there is no proposed physical mechanism for such connection without a major role of ENSO and PWC and also because the main link connecting tropical Atlantic and extratropical Pacific happens through TNA SSTAs' relationship to the pressure system over southeastern United States (assumption 3, Klein et al., 1999; García-Serrano et al., 2017; Jiang and Li, 2019). Moreover, we consider the PWC to only be connected to PNA through ENSO (e.g. via a poleward-propagating Rossby wave in the case of an El Niño event, Wallace and Gutzler, 1981; Hoskins and Karoly, 1981; Karoly, 1983), hence, we estimate the ENSO-PNA connection only through the*

*Niño3.4-PNA pair (assumption 4). Assumptions 1-4 are held throughout all results shown in Sect. 4. Additionally, during the PCMCI+ analysis of the Pacific pacemaker ensemble (Sect. 4.2.1) we assume that 5) if a contemporaneous connection is estimated between Niño3.4 and any other node, then the link should be oriented from the Niño3.4 node toward the other variable node (assumption 5) because the SSTA is nudged to observed values in the Niño3.4 region. Additionally, to avoid lagged links from PWCu to Niño3.4 due to their strong relationship, we do not estimate the influences on Niño3.4 from past PWCu (no PWCu→Niño3.4 link at any lag) in Sect. 4.2.1. Similarly, specific to the PCMCI+ analysis of the Atlantic pacemaker ensemble (Sect. 4.2.2), we assume that 6) if a contemporaneous connection is estimated between an Atlantic SST index (TNA, ATL3) and any other variable, then the link should be oriented from the Atlantic node toward the other node and the same for the direction of the contemporaneous NAO-PNA connection which is assumed as NAO→ PNA (as this was the most estimated direction when no assumption is introduced, see Supplementary Fig. S13).This is because SSTA over the TNA region is nudged toward observations, and a part of the ATL3 region is included in the linearly tapering buffer zone that extends to the equator (assumption 6). The ensemble-averaged time series (with 25th-75th percentile range shading) for all indices calculated from the Pacific and Atlantic pacemaker simulations are shown in Supplementary material Figs. S4 and S5, respectively. On another note, 7) the contemporaneous link between Niño3.4 and PWCu was detected as a conflicting link in several instances when no assumption is introduced (see odd-numbered Figs. S7-15 in Supplementary material). This might be due to the positive Bjerknes feedback loop inadequately captured on the 3-monthly averaged time resolution. Consequently, we assume this connection to be directed as PWCu → Niño3.4 (assumption 7) as this direction occurred the most frequently in the analysis of reanalysis data without any assumptions (excluding unoriented and conflicting links). Assumption 7 is not valid for the PCMCI+ analysis on the Pacific pacemaker ensemble (Sect. 4.2.1), where assumption 5 holds. Finally, to overcome specific instances of conflicting links in Sect. 4.1 between Niño3.4 and ATL3 (analysis of the observed historical period), we 8) assume the orientation of the same-sign contemporaneous Niño3.4-ATL3 connection as Niño3.4 → ATL3 (assumption 8). However, it should be noted that the relationship set by assumption 8 is fragile as discussed in Chang et al. (2006) and the same-sign effect proposed by Latif and Grötzner (2000) for ENSO's influence on Atlantic Niños was found to lag by 6 months. All causal graphs obtained without introducing any assumption are shown in odd-numbered Supplementary Figs. S7-15.*

5.      Discussion of Fig. 6b is rather hand wavy. What are the correlations between the TNA time series? ENSO should be able to impact TNA, right? This comes out from the Network, so we should see some correlation, perhaps with a 1 season lag?

**Yes, the correlations should come out in the resulting PCMCI+ networks with respect to the underlying assumptions. We understand that Fig. 6b is hard to follow and we address this point in the new manuscript by: 1) removing Figure 6 and showing the ensemble time series for the 6 variables (TNA, Nino3.4, ATL3, PNA, NAO, PWCu) in Supplementary material Fig. S4 for the Pacific pacemaker ensemble and Fig. S5 for the newly added Atlantic pacemaker ensemble. 2) The new Figs. 5-6 in Section 4.2 (pacemaker runs) now show the PDV and AMV (7-yr low-pass filtered) ensemble averaged time series (panels a and b) along with the sliding window PCMCI+ (before and after removing MEM, panels c and d correspondingly). The PDV and AMV time series plots (Fig. 5a-b, 6a-b) include a 25th-75th percentile range to**

**represent the variance between the individual pacemaker simulations. Although not included in the causal analysis, the ensemble averaged PDV and AMV time series (low-pass filtered) serve as a proxy for the long-term state of the Pacific and Atlantic and help assess the changing effect of external forcing on both basins over time in the Pacific and Atlantic pacemaker simulations.**

"Minor: Section 4.1 For which season do you calculate the regressions shown in Figs. 3 and 4 ?"

**The regressions were shown based on continuous monthly time series at lag-zero. Following the addition of ATL3 into the causal analysis, we decided to remove Sect. 4.1.1 (Correlation and regression patterns) including Figures 3 & 4 and their respective discussions. Lead/lag correlation maps have been produced extensively in previous papers (e.g. Wang, 2002; Rodríguez-Fonseca et al., 2009, Münnich and Neelin, 2005; Jimenez et al., 2021) tackling different aspects of Atlantic-Pacific interactions at different seasons and timescales. Therefore, we believe keeping such plots in the new manuscript is unnecessary and drives the paper away from the causal analysis of time series.**

**Citation**: https://doi.org/10.5194/egusphere-2023-1861-RC1

References:

Gerhardus, A. and Runge, J.: High-recall causal discovery for autocorrelated time series with latent confounders, Advances in Neural Information Processing Systems, 33, 12 615–12 625, 2020.

Karmouche, S., Galytska, E., Runge, J., Meehl, G. A., Phillips, A. S., Weigel, K., and Eyring, V.: Regime-oriented causal model evaluation of Atlantic–Pacific teleconnections in CMIP6, Earth System Dynamics, 14, 309–344, 2023.

*Smirnov, D. and Bezruchko, B.: Spurious causalities due to low temporal resolution: Towards detection of bidirectional coupling from time series, Europhysics Letters, 100, 10 005, 2012.*

Wang, C.: Atlantic climate variability and its associated atmospheric circulation cells. Journal of climate, 15(13), 1516-1536, 2002.

Rodríguez-Fonseca, B., Polo, I., García-Serrano, J., Losada, T., Mohino, E., Mechoso, C. R., and Kucharski, F.: Are Atlantic Niños enhancing Pacific ENSO events in recent decades?. Geophysical Research Letters, 36, 20, 2009.

Münnich, M., and Neelin, J. D.: Seasonal influence of ENSO on the Atlantic ITCZ and equatorial South America. Geophysical research letters, 32, 21, 2005.

Jimenez, J. C., Marengo, J. A., Alves, L. M., Sulca, J. C., Takahashi, K., Ferrett, S., and Collins, M.: The role of ENSO flavours and TNA on recent droughts over Amazon forests and the Northeast Brazil region, International Journal of Climatology, 41(7), 3761-3780, 2021.

Cassou, C., Cherchi, A., and Kosaka, Y., eds.: AR6, Annex IV: Modes of Variability, pp. 2153–2192, IPCC, Cambridge, United Kingdom and New York, NY, USA, 2021.

*Chang, P., Fang, Y., Saravanan, R., Ji, L., and Seidel, H.: The cause of the fragile relationship between the Pacific El Niño and the Atlantic Niño, Nature, 443, 324–328, 2006*

*Murtugudde, R. G., Ballabrera-Poy, J., Beauchamp, J., and Busalacchi, A. J.: Relationship between zonal and meridional modes in the tropical Atlantic, Geophysical Research Letters, 28, 4463–4466, ttps://doi.org/https://doi.org/10.1029/2001GL013407, 2001.*

*Jiang, L. and Li, T.: Relative roles of El Niño-induced extratropical and tropical forcing in generating Tropical North Atlantic (TNA) SST anomaly, Climate Dynamics, 53, 3791–3804, 2019*

*Latif, M. and Grötzner, A.: The equatorial Atlantic oscillation and its response to ENSO, Climate Dynamics, 16, 213–218, 2000.*

Newly added figures and their captions:

[Figure]

Figure 2. Causal analysis of the for the 1997-1998 El Niño event. a) Detrended standardized monthly time series of the variables listed in Table 1 between 1995-1999. b) Causal network representing lagged (curved) and contemporaneous (straight) causal links, constructed by applying PCMCI+ on the time series in (a). Nodes represent the time series associated with each climate variable (see node labels and details in Table 1). Node colors indicate the self-link strengths of each time series (auto-MCI, see color bar), and the color of the links denotes the cross-link strengths (cross-MCI, see color bar). The link-associated time lags (unit=1 month) are shown as small labels on the links.

[Figure]

Figure 3. Observed Atlantic-Pacific interactions. a) 10-year low pass-filtered AMV (solid) and PDV (dashed) from the HadISST dataset, calculated following the definition in Sect. 2.4. b) Sliding window analysis where PCMCI+ is applied for five 20-year windows moving by 10 years (see subtitles for each causal graph). In this panel (b, OBS), the algorithm is run on the original time series before removing MEM. The link-associated time lags (unit=1 season i.e 3 months) are shown as small labels on the curved links. c) Similar to (b) but using data where the MEM is removed (OBS minus MEM).

[Figure]

**Figure 4. Causal networks representing Atlantic-Pacific teleconnections for 1950-1983 (left column) vs 1983-2014 (right column) in the Reanalyses datasets. (a) Constructed by applying PCMCI+ the 1950-1983 period on the original time series before removing MEM. (b) Same as (a) but for the 1983-2014 period. (c) same as (a), but with indices calculated after removing MEM. (d) Same as (c) but for the 1983-2014 period.**

[Figure]

[Figure]

**Figure 6. Atlantic pacemaker simulations where North Atlantic SSTAs have been nudged toward observations (see Sect. 2.2). Panels a-d here are similar to Fig. 5a-d but for the Atlantic pacemaker ensemble.**

[Figure]

**Figure 7.** Regime-oriented PCMCI+ analysis of the CESM2 pre-industrial control run. a) Causal graph where only time steps corresponding to either a) PDV+/AMV+, b) PDV-/AMV+, c) PDV+/AMV-, d) PDV-/AMV- regime are considered. The regimes are defined according to the phase of e) smoothed AMV and PDV indices (13-year low-pass filtered) illustrating the decadal internal variability over the Atlantic and Pacific for 250 years (1000 time steps [seasons (3-monthly averages)]. Positive (negative) phases are shaded in pink (light blue) and red (blue) for PDV and AMV, respectively.)

**Reply to RC2**

We thank the referee for his/her constructive comments. We addressed every aspect in the comments and made a major revision to our analysis and manuscript. We enumerate the Referees' Comments in blue in "quotation marks" and we address them below in **bold**. We use the following notation: P1 L1 means Page 1, Line 1 to refer to passages from the old manuscript. We put in *italic and bold* the text to be added to the new version of the manuscript. We note that this revision takes into account major comments from the first referee (see reply RC1). The newly added Figures and their captions (and supplementary material) can be found at the very end of this document (and attached).

Main comments:

1. "The authors claim in the abstract (lines 14-16) that "causal discovery can quantify previously unknown connections and thus provides important potential to contribute to a deeper understanding of the mechanisms driving changes in regional and global climate variability". However, all along the article I didn't see clearly the adding value from the causal discovery approach used by the authors to previous knowledge."

**We thank the Referee for pointing this out. We understand that the statement might seem controversial in the given context of previously unknown connections. The statement is rectified in the new version of our manuscript to " detect previously documented connections" referring to the revised proof-of-concept section. We note that we explain the physical processes for the underlying links based on previous studies. After the adjustments made to the analysis during this revision (following both Referee comments), we updated the abstract accordingly which now reads starting L17:**

***[…] We show that causal discovery can detect previously documented connections and provides important potential for a deeper understanding of the mechanisms driving changes in regional and global climate variability.***

2. "First, in their proof of concept, the causal networks for ENSO reveals a causal link between wind stress in the western Pacific and precipitation in the Central pacific 3-month later. This link appears as independent of the central Pacific SST response to wind stress forcing. I don't know how much importance we need to accord to this link (previously unknown?): what are its physical bases? Currently, I do have the impression that this causal link is actually an artifact and that it is due to the impacts of the Wind Stress WPAC on SST Niño3.4 (2-month lag) followed by the impacts of the SST Niño3.4 on Precip CPAC (1-month lag). This makes me wonder how confident we can be regarding the detection of "previously unknown connections" with this method. Some discussion is needed to explain this point and give credit to the method."

**We updated the "Proof-of-concept" based on comments of RC1 and RC2. In the updated version of the manuscript, we replaced the precipitation anomalies (Precip central CPAC) with the eastward wind anomalies in the central Pacific (Uwind CPAC). To estimate the contemporaneous link in addition to the lagged ones, we changed the minimum time lag to 0, which helps to better capture Bjerknes feedback. We believe the updated Fig. 2 better showcases the proficiency of the method in detecting various aspects of the El Niño 1997-1998 event. We also**

acknowledge that artefacts (similar to one raised by this comment) do emerge and their interpretation should always be considered according to background knowledge and the respective uncertainties of each estimated adjacency (p-value). A matrix of p-values for every causal graph shown in the revised manuscript is now part of the supplementary material Fig. S3. More details on the confidence of the links are presented in the reply to the next comment.

3.        "Second (and third), the authors split the observed record into 2 periods (1950-1983 vs 1985-2014) and apply a causal network analysis for each of these periods. This is in fact to verify whether an already proposed (from previous studies) regime switch around 1985 in the inter-connections between Atlantic and Pacific can be identified with a causal discovery algorithm. However, I don't find this very novel nor satisfactory since the time splitting is in fact done from previously suggested period. It would have been more interesting if the author's approach was able to detect changes in behavior from a continuous timeseries. In addition, I understand that the outputs of the causal discovery analyses are not necessarily robust (as acknowledged lines 679-682). Specifically, discarding 1 year of the 1985-2014 period leads to different results (cf. differences between Figures 5bd and S1ab). This leads me to my last comment on the method: how robust are the results discussed in the article? For example, are the differences between Figures 5a and 5b meaningful accounting for the limited sample length and the independent randomness of the variables investigated? Without giving a confidence interval in the estimated coefficients of the causal networks, I don't think we can objectively interpret the results. (Similarly, how robust are the differences between raw and externally forced signal removed figures, e.g. Fig 5b vs 5d?)"

We appreciate the referee's comment and agree that a better approach can be applied to replace the splitting of the period based on previous research. Originally, the periods were relative to when AMV is trending negative vs trending positive. In the revised analysis, we address this comment by applying a sliding window analysis using PCMCI+ (see e.g. Fig. 3 for reanalysis in the revised manuscript). This is done by running the algorithm for 5 windows of 80 seasons (3-monthly averages) each i.e. 20 years. We start at 1950 and move the window by 10 years. The results are networks for each of the following periods: 1950-70, 1960-80, 1970-90, 1980-2000 and 1990-2010. We find this approach to better showcase the changing interactions over time and fits better to the scope of the paper. Apart from reanalysis data, such analysis is shown for the pacemaker runs in the revised manuscript (Sect. 4.1 and Sect. 4.2). Additionally, the causal analysis of reanalysis data (Sect. 4.1) now also includes plots for 1950-1983 vs 1983-2014 causal graphs (this time without disregarding any year, Fig. 4). It is noteworthy that methods tailored to detect regime-dependent causal relations from time series are available (Saggioro et al., 2020) but not suitable to current high dimensionality and the short sample sizes of our data frames. Concerning the independence tests employed, the statistical significance of the results and related limitations, the methodology section of our revised manuscript now reads starting L193:

*[...] There are two main free parameters for PCMCI+. First, the maximum time lag $\tau_{max}$, which is decided after analyzing the lagged dependencies between the variables (see lag function plot in supplementary Fig. S2). The second parameter is $\alpha_{pc}$, which represents the significance threshold adopted for all PCMCI+ tests. The algorithm outputs a p-matrix (containing the p-values, denoting the uncertainty of each link) and a val_matrix (containing the cross-MCI, translating the strength of each link). The p-values for the coefficients shown on a PCMCI+ causal graph are below the*

*significance threshold α$_{pc}$. We note that these are valid only for the adjacencies and not the directionality of contemporaneous links decided during the orientation phase (lagged links are always oriented according to time order). The limitation presented by the absence of comprehensive confidence measures for all PCMCI+ estimated links is an aspect currently being addressed where bootstrap aggregation methods are still under review (Debeire et al., 2023, in review). In the context of this paper, the p-value matrices are shown in Supplementary Material (e.g. Figs. S3, S6, S8). to provide a measure of the uncertainties of all estimated adjacencies.*

4.      "The authors use a 120-year long piControl simulation performed with CESM2 to investigate the impacts of different background state in the Pacific and Atlantic on their interannual interactions. To do so, they split those 120 years into 3 periods of 40 years and perform a causal discovery analysis for each of them. They found differences and claim it is controlled by the background changes. Again, this analysis need to be done considering error on the estimates of the coefficients. Are those differences meaningful? The authors could have use more than 120 years of this piControl simulation and could have actually addressed this question computing those coefficients for every possible 40-year windows of the simulation. And then use a composite approach based on the states of the PDV and AMV, verifying whether the coefficients from those different composite pools were statistically different."

**We considered the referee's recommendation and adjusted our analysis of the pre-industrial run. The revised manuscript analyses a longer period of 250 years (1000 time steps i.e. seasons) using a regime-oriented approach depending on the in-phase and out-of-phase combination of the 13-year low-pass filtered AMV and PDV. The composites for each regime are constructed by filtering out the time steps for each combination: PDV+/AMV+, PDV-/ AMV+, PDV+/AMV-, and PDV-/AMV-. The low-pass filtered PDV and AMV are only used for identifying the time steps to be considered for PCMCI+ and are not part of the causal analysis data frame. The p-value matrices for the pre-industrial results are also now included in the supplementary material in Fig. S14. We add this in P12 L301:**

*Concerning Sect 4.3, we show results based on composites selected depending on the phase combination of PDV and AMV (i.e. PDV+/AMV+, PDV-/AMV+, PDV+/AMV- and PDV-/AMV-). A mask is used on the PCMCI+ data frame to only select time steps that satisfy a certain combination. A similar "regime-oriented" analysis is detailed in Fig. 3 in (Karmouche et al., 2023). Here, we only use this approach for the pre-industrial control run because larger sample sizes are available and such sampling at lower intervals on the short reanalysis and pacemaker data might produce "spurious" results (Smirnov and Bezruchko, 2012).*

5.      The authors apply a causal network on monthly (or 3-month mean) indices without considering possible changes in the interplay between those indices among different seasons. However, it has been proposed, for example, that DJF NIÑO3.4 SST anomalies are creating TNA SST anomalies of the same sign in the following MAM season, and that JJA TNA SST (as well as JJA ATL3 SST) anomalies are creating NIÑO3.4 SST anomalies of opposite sign in the following DJF season. Without accounting for this seasonality in the interactions, I believe the linked coefficients reported in the current article are actually a mixed between those interactions. Therefore, I am inviting the authors to include this seasonality in their analyses. I believe this could be done by splitting each node in 4 quadrants representing the 4 seasons. (Possibly, the lagged auto-correlation of those

indices could also be represented with arrows between those quadrants, accounting for the seasonality of this auto-correlation…).

We thank the Referee for pointing this out. We do recognize that the different modes analysed have different peak seasons and hence the strength of the causal relationships are also season-dependent. We also believe that, although we are using complete-year time series (3-monthly averaged), the method is able to correctly estimate the sign and the lag of the dominant causal links during the analysed period. The coefficients indeed quantify the overall effect over all considered timesteps, but the dominant (and most significant under $\alpha_{pc}$) effects are the ones that are shown on the causal graph. We recognize that the approach proposed by the referee would better quantify the causal effects of single seasons, however, the splitting into 4 quadrants is not suitable due to sample size constraints. At least for the reanalysis and pacemaker ensembles, splitting each node to 4 different seasons would dramatically shorten the data frame and impair the causal analysis. We therefore address this constraint in the revised methodology section in P12 L301 (Sect. 3.2.2) by adding the following:

*We also recognize the potential benefits of a more granular analysis, such as splitting each time series into four quadrants (representing four seasons) to analyze the effects on a particular season (e.g. on the peak season of a certain variable or summer vs winter periods). However, implementing such a quadrant-based method would dramatically shorten the data frame and impair the causal analysis, especially for reanalysis and pacemaker ensembles (where short periods are analyzed). Despite this limitation, the adopted methodology which considers the complete-year time series (3-monthly averaged) identifies the overall effect considering all time steps (seasons) and provides valuable insights into the seasonal lag and strength of causal relationships. Naturally, the most significant links (under the $\alpha_{pc}$ threshold) are the ones that come out on the causal graph, and which should always be interpreted according to background knowledge (e.g. with respect to the peak season of the variables analyzed).*

6.     "The authors are using pacemaker simulations in which SST is forced over Central-Eastern tropical Pacific to investigate causal relationship. I don't understand how those kind of simulations can be useful to investigate the influence of the Atlantic on the Pacific as the Pacific SST are constrained by construction and are therefore not responding to potential Atlantic forcing. The authors should comment on that. In the same line, the causal networks of those experiments show that the PWCu is causing changes to the Nino34 SST. Could the authors explain how this is possible whereas the Nino34 SST is actually imposed?"

We thank the Referee for rising this issue. Similar comment was rised by Referee 1 comment #4. To give credit to the method, the issue of the earlier pacemaker results stems from the post processing of the PCMCI+ graphs. In the old manuscript, we transformed the original cross-MCI values (link strengths) to standardized linear link coefficients estimated after fitting a linear mediation model to the set of parents estimated by PCMCI+. The idea behind this was to better quantify the causal links by providing a measure of the change in units of standard deviations. The challenge that rises with this approach is that it requires a fully oriented Directed Acyclic Graph (DAG), which is not always the case for the resulting PCMCI+ graphs, as they might contain unoriented links at lag-zero, either as contemporaneous unortiented adjacencies (denoted as 'o-o') or conflicting links (represented by 'x-x').

Transforming such links (exclusive to lag-zero) was based in most cases on a majority ruling among the CESM2 pacemaker ensemble (as previously explained in P8 L201-207). We recognize this was not an ideal approach especially when it comes to providing the ensemble average graphs. The method originally detected several unoriented links between Nino3.4 and PWC in the individual pacemaker simulation runs (compared to both observations and pre-industrial control) but those undirected links were not taken into account for deciding the overall direction in the transformation to fully oriented DAG. To address these issues we revised our analysis by: 1) showing the original cross-MCI values (instead of the standardized link coefficients) for expressing link strengths. This is in order to tolerate links that denote contemporaneous adjacencies ('o-o'). In the remaining cases of conflicting links ('x-x'), we 2) introduce background knowledge as assumptions based on sensitivity tests and/or well-established physical processes. These assumptions are used to address the main issue presented by this referee comment. The Pacific pacemaker ensemble causal analysis is now supported now by a background assumption that states: in the case of a detected lag-zero adjacency between the nudged variable (i.e. Nino3.4) and another variable, the link should be directed from the nudged variable to the other variable. The added paragraphs to the new manuscript can be found part of our reply to comment #4 from Referee 1.

Minor comments:

- "reference for Walker Circulation (Bjeknes, 1969) is introduced line 42 instead of line 35."

Changed position to L35. Thank you.

- "line 62: "While debate over the precise attribution of recent warming trends", the authors may be referring to the global warming hiatus of the 2000-2014 period, but I am not completely sure. Please clarify."

Yes, we originally referred to the 2000-2014 warming slowdown compared to the previous warming period 1971-2000. We clarify this and change the statement to the following:

*While debate over the precise attribution of the early-2000s warming slowdown…*

- "line 68: "differences in the considered timescales", I don't understand to what this is related. Remove?"

Removed.

- "Section 2.4: NAO indices defined from seasonal mean EOF. I am surprised by this choice as the locations of the NAO centers of action are not the same between the 4 seasons. Given the higher variance of the winter SLP, it is likely that the EOF is capturing the winter NAO pattern. I am then wondering the physical meaning of this for the other seasons, especially in summer. This comment could also apply for the PNA index."

This is to stay consistent with the 3-monthly averaged data resolution across all variables. The applied PCMCI+ algorithm  requires all time series to have the same

**length. This point has been discussed in more detail in our reply to comment #5 in the current RC2.**

    - lines 128 and 130: please defined what you mean exactly by "seasonally averaged", i.e. 3-month.

**In L128 and L130 we changed "seasonally" to "3-monthly". We note that during the revision analysis we also adjusted the seasons to DJF, MAM, JJA, SON (instead of JFM, AMJ, JAS, OND)**

    - line 132: "for the pacemaker simulations", I believe this comment is valid for the piControl simulation too.

**The statement is only valid for the pacemaker simulations. The eastward wind data for piControl is available at the 925 hPa pressure level.**

    - lines 139-144: definition of the AMV and PDV indices. The variations of the global averaged SST index are driven by external forcing but also by the expression of regional internal variability. For example, if the SST over the whole globe were showing no anomaly except in the North Atlantic, the global SST average anomaly would be due to the North Atlantic. By removing the global SST average to local SST, one not only "detrend" the local data, but also remove the global expression of the regional variability. I understand that this approach might be used in same circumstances. However, in the present article the authors are estimating the effect of external forcing from a CMIP6 model multi-ensemble mean (cf. Section 3.1), detrending the observed data with it. Therefore, I would advice to not remove the global SST average to estimate the AMV and PDV indices.

**We follow the referee's advice and remove "minus the global mean (70°N–60°S, effectively detrending the data)" in L140 and L144. We remind that we don't use the AMV and PDV as part of the PCMCI+ data samples. The low-pass filtered PDV and AMV indices presented in the figures of the revised manuscript are calculated to represent raw anomalies without subtracting the global mean SST. We also show the low-pass filtered PDV and AMV indices with the multi-ensemble mean (MEM) subtracted in Sect. 4.2 (pacemaker). The smoothed time series serve as indicator of the long-term state of the Pacific and Atlantic. As mentioned earlier, PDV and AMV are also used in the regime-oriented analysis of Sect. 4.3 (piControl) to filter out the time steps to be considered by PCMCI+.**

    - line 162: "Consequently, the discrepancies in each pacemaker simulation relative to MEM can be attributed to internal variability", please note that the differences between the CESM2 pacemaker simulations and the MEM can also come from: (i) the real climate sensitivity to external forcings in the restoring region being different from the one estimated by the MEM, (ii) the specific climate sensitivity of CESM2 to external forcings (i.e. externally forced signal in CESM2 is different from MEM) outside of the restoring region and, (iii) a mixed between (i) and (ii).

**We thank the referee for this comment. We address this issue in our revised manuscript in L169 as follows:**

*It should be noted that the differences between the pacemaker simulations and MEM can emerge from: i) MEM estimating a climate sensitivity (in response to external forcing) different from the one prescribed in the restoring region. ii) Outside the restoring region, the climate sensitivity of the CESM models can also be different from the one estimated by MEM there. iii) Differences can also arise from a mix of (i) and (ii).*

**We also change "(producing an isolated internal variability)" in L175 to "(producing an estimate of isolated internal variability)"**

- line 262-263: "The changes in precipitation patterns further affect the atmospheric and oceanic conditions, reinforcing the warming in the Niño3.4 region". This does not appear on Figure 2, remove? Or at least makes it clear that it is an information capture from the causal network but coming from literature.

**That is now removed as part the revision regarding comment #2 in RC1.**

- Figures 2, 5, 7 and 8: why not using the same scale for "auto-coeff." and "link coeff."? Are those coefficient of different nature? If it is the case, please make it clearer in the method section, if not, I would advice to use only one color bar. Also, please use the same expression in all those figures. In the current state, some figure labels are saying "coeff." whereas other are saying "corr." and "strength".

**Please excuse the oversight. In the causal graphs of our revised manuscript (which now show the originally estimated cross-MCI values instead of the linear "link-coeff") we display a standard color bar with the same scale and labels.**

- line 509: please substitute "two members in each experiment" by "two members in each period".

**Substituted.**

- line 509: please substitute "in the four experiments" by "in the four cases".

**Substituted.**

- lines 534-535: "small differences, most likely originating from observational uncertainty, ERSSTv5 vs HadISST". The differences can also be due to the efficiency of the SST restoring, since the SST are not imposed as in atmospheric only simulation but they are nudged.

**We note that Figure 6 would be moved to Supplementary material as Fig. S4 as part of the revision following comment #5 in RC1 (this is for time series of all variables for both pacemaker ensembles, before and after subtracting MEM). Hence, any related discussion of Fig. 6 of the old manuscript would be removed from the revised text.**

- lines 535-536: "The pacemaker ensemble mean TNA (red lines in Fig. 6b) implies an important role of ENSO in shaping SSTAs over the Atlantic". I don't see the basis for this comment, could you explain why you consider that ENSO play an important role in shaping SSTAs in the Atlantic from Figure 6b?

**Our response to the previous comments also addresses the raised issue.**

> - line 592-593: "The warming SST trends in the Atlantic favor a strengthened PWC which ultimately cools the SSTs over the Niño3.4 due to enhanced upwelling". According to P3 in igure 8b, this is happening with a 1-year lag. What physical process can explain that tropical SST anomalies are changing tropical atmospheric circulation 1 year later?

**This is now changed following the adoption of the AMV/PDV phase-dependent approach for the piControl section. In the revised version, the effect of the Atlantic on the Pacific in the piControl run is detected from ATL3 (and not TNA; with $\alpha_{pc}$ = 0.05) to PWCu with 1-season lag. We also highlight that the maximum time lag parameter has been set to 4 instead of 8 seasons following comment #3 from Referee 1.**

> - lines 601-605: how meaningful are those changes? See my main comments 1 and 2.

**We understand this comment is related to the uncertainty of the coefficients. In our revised manuscript, p-value matrices for the new piControl results are included in the supplementary material. See also replies to main comments #1 and #2.**

References

Saggioro, E., de Wiljes, J., Kretschmer, M., & Runge, J.: Reconstructing regime-dependent causal relationships from observational time series. Chaos: An Interdisciplinary Journal of Nonlinear Science, 30(11), 2020.

Newly added Figures and their captions:

[Figure]

**Figure 2. Causal analysis of the for the 1997-1998 El Niño event. a) Detrended standardized monthly time series of the variables listed in Table 1 between 1995-1999. b) Causal network representing lagged (curved) and contemporaneous (straight) causal links, constructed by applying PCMCI+ on the time series in (a). Nodes represent the time series associated with each climate variable (see node labels and details in Table 1). Node colors indicate the self-link strengths of each time series (auto-MCI, see color bar), and the color of the links denotes the cross-link strengths (cross-MCI, see color bar). The link-associated time lags (unit=1 month) are shown as small labels on the links.**

[Figure]

**Figure 3. Observed Atlantic-Pacific interactions. a)** 10-year low pass-filtered AMV (solid) and PDV (dashed) from the HadISST dataset, calculated following the definition in Sect. 2.4. **b)** Sliding window analysis where PCMCI+ is applied for five 20-year windows moving by 10 years (see subtitles for each causal graph). In this panel (b, OBS), the algorithm is run on the original time series before removing MEM. The link-associated time lags (unit=1 season i.e 3 months) are shown as small labels on the curved links. **c)** Similar to (b) but using data where the MEM is removed (OBS minus MEM).

[Figure]

**Figure 4.** Causal networks representing Atlantic-Pacific teleconnections for 1950-1983 (left column) vs 1983-2014 (right column) in the Reanalyses datasets. (a) Constructed by applying PCMCI+ the 1950-1983 period on the original time series before removing MEM. (b) Same as (a) but for the 1983-2014 period. (c) same as (a), but with indices calculated after removing MEM. (d) Same as (c) but for the 1983-2014 period.

[Figure]

[Figure]

**Figure 6. Atlantic pacemaker simulations where North Atlantic SSTAs have been nudged toward observations (see Sect. 2.2). Panels a-d here are similar to Fig. 5a-d but for the Atlantic pacemaker ensemble.**

[Figure]

**Figure 7. Regime-oriented PCMCI+ analysis of the CESM2 pre-industrial control run. a) Causal graph where only time steps corresponding to either a) PDV+/AMV+, b) PDV-/AMV+, c) PDV+/AMV-, d) PDV-/AMV- regime are considered. The regimes are defined according to the phase of e) smoothed AMV and PDV indices (13-year low-pass filtered) illustrating the decadal internal variability over the Atlantic and Pacific for 250 years (1000 time steps [seasons (3-monthly averages)]. Positive (negative) phases are shaded in pink (light blue) and red (blue) for PDV and AMV, respectively.)**